# Structural and mechanistic basis for redox sensing by the cyanobacterial transcription regulator RexT

Bin Li[1], Minshik Jo[1], Jianxin Liu [1], Jiayi Tian[1], Robert Canfield[1,2] & Jennifer Bridwell-Rabb [1✉]

Organisms have a myriad of strategies for sensing, responding to, and combating reactive oxygen species, which are unavoidable consequences of aerobic life. In the heterocystous cyanobacterium *Nostoc* sp. PCC 7120, one such strategy is the use of an ArsR-SmtB transcriptional regulator RexT that senses $H_2O_2$ and upregulates expression of thioredoxin to maintain cellular redox homeostasis. Different from many other members of the ArsR-SmtB family which bind metal ions, RexT has been proposed to use disulfide bond formation as a trigger to bind and release DNA. Here, we present high-resolution crystal structures of RexT in the reduced and $H_2O_2$-treated states. These structures reveal that RexT showcases the ArsR-SmtB winged-helix-turn-helix fold and forms a vicinal disulfide bond to orchestrate a response to $H_2O_2$. The importance of the disulfide-forming Cys residues was corroborated using site-directed mutagenesis, mass spectrometry, and $H_2O_2$-consumption assays. Furthermore, an entrance channel for $H_2O_2$ was identified and key residues implicated in $H_2O_2$ activation were pinpointed. Finally, bioinformatics analysis of the ArsR-SmtB family indicates that the vicinal disulfide "redox switch" is a unique feature of cyanobacteria in the *Nostocales* order, presenting an interesting case where an ArsR-SmtB protein scaffold has been evolved to showcase peroxidatic activity and facilitate redox-based regulation.

[1] Department of Chemistry, University of Michigan, Ann Arbor, MI, USA. [2] Present address: Department of Microbiology and Immunology, University of Colorado Anschutz Medical Campus, Aurora, CO, USA. ✉email: jebridwe@umich.edu

The diversity of environments that photosynthetic microorganisms are equipped to survive in is unparalleled. These organisms inhabit environments that contain different amounts of molecular oxygen ($O_2$) and experience different wavelengths and intensities of light. To thrive in these environments, photosynthetic organisms must have strategies to perceive and defend against the production of reactive oxygen species (ROS), a class of $O_2$-derived species that includes superoxide ($O_2^{·-}$), peroxide ($O_2^{2-}$), hydroxyl radicals ($OH^{·-}$), singlet oxygen ($^1O_2$), and hydrogen peroxide ($H_2O_2$)[1,2]. ROS are generally referred to as the inescapable cost of aerobic metabolism and are known to arise in photosynthetic organisms from the continual liberation of molecular oxygen ($O_2$), the electron transport chains of aerobic respiration and photosynthesis, and via reaction of excited state photosynthetic pigments with $O_2$[2]. As with all organisms, the production of ROS in photosynthetic organisms can have detrimental effects. ROS react with metal ions and perpetuate their own formation, and cause oxidative damage to protein, DNA, and lipid biomolecules found in cells[1,2]. To circumvent ROS-linked damage, photosynthetic organisms contain elaborate defense mechanisms for detoxification. For example, small molecule antioxidants and a wide array of antioxidant enzymes, including superoxide dismutase, catalase, peroxiredoxin, and rubrerythin are used to combat ROS[3,4]. In addition, many photosynthetic organisms use thioredoxin proteins to reduce disulfide bonds that are formed in response to ROS[5]. Finally, a whole host of regulatory proteins exist to initiate signaling cascades in response to increased levels of ROS[6].

The paradigm $H_2O_2$ sensor is the LysR-type transcriptional regulator from Gram-negative bacteria known as OxyR[7,8]. This protein senses the presence of $H_2O_2$ via formation of an intramolecular disulfide bond[8]. Dependent upon the bacterial OxyR homolog, it is known that a resultant conformational change either triggers its ability to activate or de-repress transcription of genes involved in ROS protection[8,9]. Despite the utility of OxyR in bacteria for responding to increases in $H_2O_2$ concentrations, there is not a known OxyR ortholog in photosynthetic cyanobacteria. Recently, however, a few details into how $H_2O_2$ stress is

sensed in the heterocystous cyanobacterium *Nostoc* sp. PCC 7120 (also referred to as *Anabaena*) have been revealed. This organism is a filamentous photosynthetic cyanobacterium from the *Nostocales* order[10] which can differentiate into vegetative cells and microoxic compartments, known as heterocysts, where nitrogen fixation occurs. These organisms have received attention due to the possibility of engineering other photosynthetic organisms to fix nitrogen or produce hydrogen in the heterocyst environment[11,12]. For this endeavor, understanding how to control the production of ROS is paramount as ROS can damage the complex metallocenters of nitrogenase and hydrogenase. On the other hand, these heterocyst-forming organisms have also garnered interest due to their participation in harmful algal bloom formation in nitrogen-depleted environments[13]. Here, uncontrolled ROS production is beneficial as increased concentrations of ROS have shown promise in mitigating the propagation of blooms[14]. In both cases, there is clearly a need to understand, at the molecular level, how these organisms' sense, control, respond to, and adapt to ROS.

For *Nostoc* sp. PCC 7120, it is known that thioredoxin plays a critical role in the oxidative stress response[15-17]. In particular, transcription of the gene that encodes thioredoxin A2 (TrxA2) is induced upon exposure to $H_2O_2$ by the redox-sensing transcriptional regulator RexT[17], an annotated member of the ArsR-SmtB family of transcriptional regulators. This protein family was named after the founding members, *E. coli* arsenic and antimony regulatory protein (*Ec*ArsR)[18] and *Synechococcus elongatus* PCC 7942 $Zn^{2+}$-dependent regulatory protein (*Se*SmtB)[19], and many of its members are recognized as metal-responsive transcriptional regulators (Fig. 1a). However, there are also members in this protein family that deviate from the metal-sensing role and instead have been shown to sense ROS or reactive sulfur species[20,21]. For example, as observed with the OxyR transcriptional repressor homologs[9], RexT binds DNA under reducing conditions and dissociates from DNA following treatment with $H_2O_2$[17] (Fig. 1b). The latter dissociation step permits transcription of *trxA2* and is suggested to be caused by intramolecular disulfide bond formation between two of the three RexT Cys

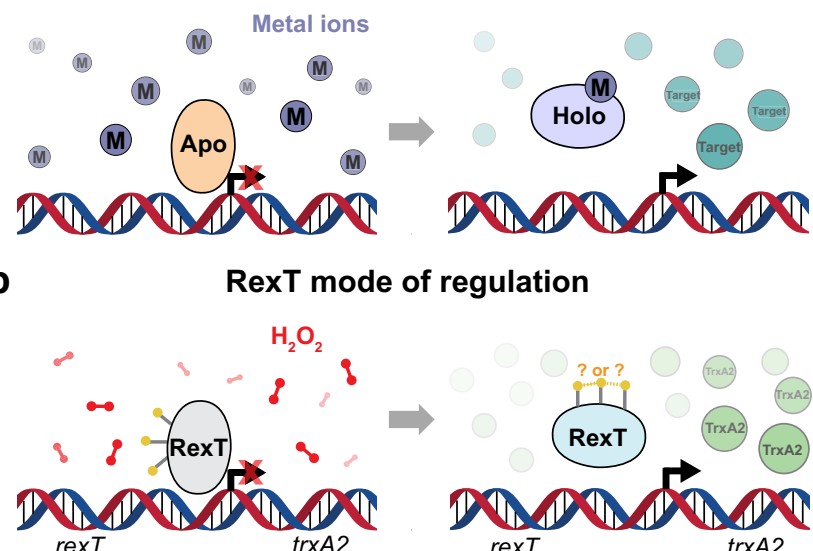

### a    Metal-binding ArsR-SmtB mode of regulation

### b    RexT mode of regulation

**Fig. 1 The ArsR-SmtB family of transcriptional regulators act as repressors of their target genes. a** The metal-binding members of the ArsR-SmtB class act as repressors of the target gene in the apo-state. In contrast, when bound to a metal ion, a conformational change ensues, and transcription of the target gene is initiated[23,24]. **b** RexT is a non-canonical member of the ArsR-SmtB family of transcriptional regulators. Rather than binding metal ions, RexT is proposed to respond to $H_2O_2$ stress through formation of a disulfide bond that initiates transcription of the thioredoxin-encoding gene *trxA2*[17].

residues (Cys40, Cys41, and Cys105)[17]. Once produced, TrxA2 can reduce disulfide bonds in target proteins and restore the ability of RexT to bind DNA via disulfide bond reduction, meaning that RexT is a reversible sensor of oxidative stress[17]. This ability of RexT to sense and respond to ROS and contribute to the maintenance of cellular redox homeostasis has led to its identification as a potential target for promoter development in applications aimed at metabolically engineering cyanobacteria to fix nitrogen or produce hydrogen[22]. However, there are still many missing details regarding how it functions. For example, it is unknown how RexT binds DNA, which residues RexT uses to sense and react to $H_2O_2$, and how the formation of a disulfide bond results in de-repression of transcription (Fig. 1).

To investigate the metal-independent regulatory mechanism by which RexT senses and propagates a response to $H_2O_2$, we used X-ray crystallography, site-directed mutagenesis, mass spectrometry, fluorescence anisotropy, and electrophoretic mobility shift assays (EMSA) to establish the molecular details of the RexT-based mode of DNA regulation. We identified key residues involved in DNA binding and pinpointed residues involved in the binding and activation of $H_2O_2$ and the formation of the regulatory disulfide bond. In addition, we established that the RexT-based mode of regulation is ubiquitous among organisms in the *Nostocales* order. Importantly, this mechanism expands the known regulatory mechanisms for sensing $H_2O_2$ and adds an extra layer of complexity to the ArsR-SmtB family as it shows that a common ancestral protein scaffold can evolve to showcase peroxidatic activity and respond to oxidative stress or sense metal ions.

## Results

**RexT has a dimeric ArsR-SmtB-like helix-turn-helix fold but lacks metal-binding motifs.** The crystal structure of RexT was determined in two stages. First, a lower-resolution 2.50-Å resolution structure of SeMet labeled RexT was solved using single-wavelength anomalous dispersion (SAD) phasing. Second, this SeMet structure was used as a model for the molecular replacement to determine a higher resolution structure of native RexT to 1.95-Å resolution (Table 1). The asymmetric unit of the RexT crystal structure contains a homodimer of RexT, which is consistent with the oligomeric state observed in solution using size-exclusion chromatography (Fig. 2a, Supplementary Fig. 1a). Each monomeric unit of RexT contains five α-helices and a pair of antiparallel β-strands, which is comparable to the α1-α2-α3-α4-β1-β2-α5 topology that is typical of the ArsR-SmtB superfamily of transcription factors[23,24] (Fig. 2a, b). The dimer interface of RexT is formed by a coiled-coil interaction between the α1 helices (residues 13–22) and between the $α5_b$ helices (residues 91–96) of each protomer (Fig. 2a, b and Supplementary Fig. 1b). The buried interface area is 707.6 Å$^2$, ~11.5-percent of the total surface area of each protomer (Supplementary Fig. 1b). The α3 helix (residues 41–43), which houses two of the three aforementioned Cys residues (Cys40 and Cys41), the α4 helix (also known as the recognition helix, $α_R$, residues 50–62), and the loop connecting them form the helix-turn-helix motif, whereas the antiparallel β-strands (residues 66–71 and 74–79) form the "wing" of the winged helix-turn-helix (wHTH) architecture[25] (Fig. 2a, b). The third Cys residue, Cys105, is located near the flexible C-terminus of the protein. This region is disordered in both monomeric units. As such, Cys105 is missing from chain A and is the last modeled residue in chain B.

A search of the protein data bank using the DALI server[26] identified the $Cd^{2+}$-binding regulator from *Staphylococcus aureus* (*Sa*CadC)[27], the ArsR homolog from *Acidithiobacillus ferrooxidans* (*Af*ArsR)[28], and two HlyU regulators proposed to be involved in redox-based regulation of virulence genes[29,30], as the

closest related structural homologs to RexT (Supplementary Fig. 2a–d). Pairwise superposition of these structures and that of *Se*SmtB[31] (Supplementary Fig. 2e) with RexT shows that the core wHTH architecture is well conserved but some structural deviations are apparent in the flexible N- and C-terminal domains (Supplementary Fig. 2). Differences are also seen in the α5 helix of RexT, which is split into two segments by a three-amino-acid turn (Phe-Pro-Gly), and in the α3 helix of RexT, which consists of only three amino-acids (Fig. 2b). Most noticeably, RexT lacks the metal-binding motifs typical of this superfamily, including the N-terminal Cys residue and the conserved ELCVCD motif on the α3 helix for binding $Cd^{2+}$ observed in *Sa*CadC[27] (Supplementary Fig. 2a, c). Likewise, RexT does not contain the three Cys binding motif on the C-terminus found in *Af*ArsR[28], or the CXCXXC motif on the α3 helix found in other ArsR homologs[24] (Supplementary Fig. 2a, d). RexT also lacks the Asp/Glu and His-rich motif that resides on the α5 helix for binding $Zn^{2+}$ found in *Se*SmtB[19,31] ($DXHX_{10}HXXE$, Supplementary Fig. 2a, e). Consistent with these observations, an EMSA was used to probe the effect of As (III), $Cd^{2+}$, and $Zn^{2+}$ on the ability of RexT to bind DNA (Supplementary Fig. 3a, Supplementary Table 1–2). As expected, based on the structure of RexT and the lack of the typical residues involved in metal ion binding, none of these ions, even at a five-fold excess, caused changes in DNA binding. A similar trend was also observed in a fluorescence anisotropy experiment, which again revealed no significant changes in DNA binding affinity in the presence of $Cd^{2+}$ (Supplementary Fig. 3b). Last, despite a similar proposed role of sensing oxidative stress, RexT also lacks the proposed functional Cys residues found on the α2 helix of HlyU (Supplementary Fig. 2a, b)[29].

**Identification of RexT structural features involved in DNA binding.** In the dimeric structure of RexT, the wHTH region

### Table 1 Data collection and refinement statistics.

| | SeMet-RexT$^§$ | RexT | RexT_ox |
|---|---|---|---|
| **Data collection** | | | |
| Space group | P6$_4$ | P6$_4$ | P6$_4$ |
| Cell dimensions a, | 100.46, | 99.89, | 100.46, |
| b, c (Å) | 100.46, 36.05 | 99.89, 36.47 | 100.46, 36.05 |
| α, β, γ (°) | 90, 90, 120 | 90, 90, 120 | 90, 90, 120 |
| Resolution (Å) | 33.3-2.50 | 33.6-1.95 | 25.1-2.16 |
| $R_{merge}$ | 0.157 (1.193) | 0.150 (1.045) | 0.104 (1.431) |
| I/σ | 12.78 (2.10) | 18.44 (2.25) | 19.44 (1.95) |
| Completeness (%) | 99.7(99.1) | 97.6 (85.5) | 99.9 (99.7) |
| Redundancy | 8.8 (8.7) | 13.7 (9.7) | 12.9 (12.8) |
| CC1/2 | 0.998 (0.718) | 0.999 (0.697) | 0.999 (0.712) |
| **Refinement** | | | |
| Resolution (Å) | 2.50 | 1.95 | 2.16 |
| Unique reflections | 7402 | 15063 | 11401 |
| $R_{work}$/$R_{free}$ | 0.229,0.258 | 0.179, 0.222 | 0.200, 0.256 |
| No. atoms | 1679 | 1897 | 1731 |
| Protein | 1616 | 1679 | 1629 |
| Glycerol | 0 | 42 | 0 |
| Chloride | 4 | 3 | 0 |
| Water | 59 | 173 | 102 |
| **B-factors** | | | |
| Overall | 47.72 | 27.33 | 50.13 |
| Protein | 47.38 | 25.95 | 50.02 |
| Glycerol | / | 47.16 | / |
| Chloride | 57.31 | 35.17 | / |
| Water | 47.77 | 35.85 | 51.77 |
| $H_2O_2$ | / | / | 62.52 |
| R.m.s. deviations | | | |
| Bond lengths (Å) | 0.012 | 0.010 | 0.009 |
| Bond angles (°) | 1.71 | 1.19 | 1.15 |

Values in parentheses are for the highest-resolution shell.
$^§$Bijvoet pairs were not merged during data processing.

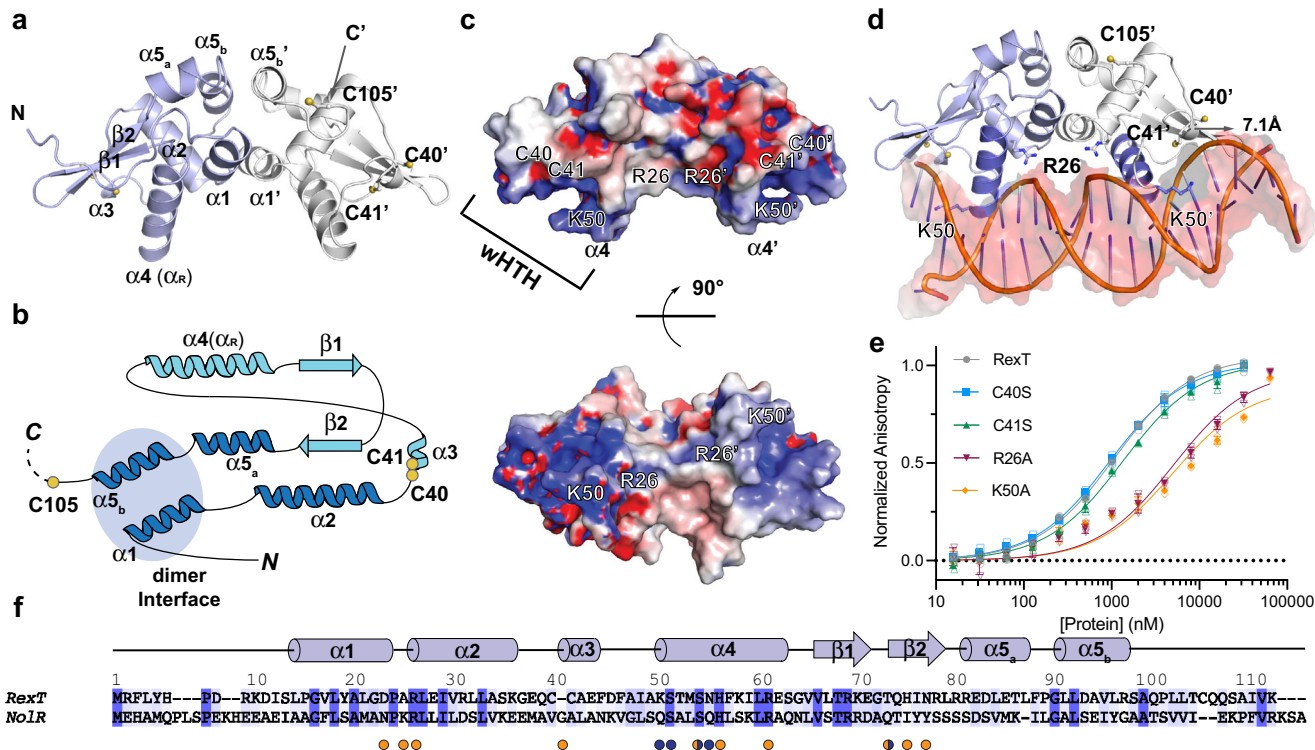

**Fig. 2 RexT has an ArsR-SmtB-type winged helix-turn-helix fold. a** The crystal structure of RexT reveals that it is a homodimer. Each protomer of RexT has an α1-α2-α3-α4-β1-β2-α5$_{a/b}$ architecture. Chains A and B are colored in light blue and light gray, respectively. Two of the three potential redox sensing Cys residues are located on the α3 helix (Cys40 and Cys41). Due to the dynamic nature of the residues following Cys105 in chain B, residues 106–112, like residues 102–112 in chain A were unable to be included in the final model of RexT. **b** A topology diagram of RexT shows the position of the three Cys residues (yellow circles), the dimeric interface (blue circle), and the wHTH motif (light blue secondary structure). **c** The calculated electrostatic potential of RexT reveals positively charged patches. The two-Cys residues and two positively charged residues that are proposed to be involved in DNA-binding are highlighted. **d** RexT is shown overlaid with a DNA-bound NolR structure[32] (PDB: 4ON0, RexT is shown with DNA from the NolR structure following an alignment performed in PyMol). This overlay reveals potential residues involved in interacting with DNA and showcases that a complementary interaction can be formed between the positively charged region of RexT and the negative backbone of DNA. In each monomer, α$_R$ is colored in dark purple. **e** A labeled DNA probe shows changes in fluorescence anisotropy following the addition of RexT. These differences allowed for the calculation of the DNA binding affinity of wild-type RexT and the different variants used in this work (Supplementary Table 2). Compared to wild-type RexT (gray, $K_d = 1.08 \pm 0.07 \mu M$), the K50A (orange, $K_d = 5.29 \pm 1.86 \mu M$) and R26A (red, $K_d = 5.18 \pm 1.22 \mu M$) RexT variants show decreased DNA binding affinity whereas the C40S (blue) and C41S (green) variants don't have any significant change. **f** A sequence alignment of RexT with NolR shows a conserved DNA-binding architecture and key residues. The sequences are aligned by Clustal W.[89] and colored by Jalview[90] based on the percent identity with dark blue indicating sequence identity. DNA binding residues in NolR are indicated by orange (interaction with DNA backbone phosphate) and blue (base-specific interaction) dots. The alignment is annotated based on the secondary structure and residue numbers of RexT. In **e**, data was measured using $n = 3$ independent experiments and is presented with the individual measurements (open shapes) and as the mean value of these measurements ± SD (closed shapes). Source data are provided as a Source Data file.

shows an overall positive electrostatic potential and appears well-poised to interact with DNA (Fig. 2c). An overlay with a related DNA-bound ArsR-SmtB member NolR[32], which exhibits a DNA binding competent conformation that is unusual for the protein family, reveals that a complementary interaction can be formed between the overall positive electrostatic potential region of RexT and the overall negative region of the DNA backbone (Fig. 2c, d). Specifically, both α4 helices (α$_R$) of the dimer are comparably spaced relative to those in NolR and are positioned to interact with the major groove of DNA, and both antiparallel "wings" are situated to interact with the DNA minor groove (Fig. 2d and Supplementary Fig. 4)[32,33]. In the structure of NolR, binding interactions with the DNA phosphate backbone are largely contributed by residues from the α2 and α3 helices as well as the β wing, whereas base-specific interactions mostly come from the α4 helix (Fig. 2d, f)[32]. For example, Arg31 on the α2 helix of NolR interacts with the phosphate group on the DNA backbone[32]. Changing this residue into an alanine residue was demonstrated to abolish the ability of NolR to interact with DNA using

isothermal titration calorimetry[32]. The corresponding residue in RexT is Arg26 (Fig. 2f). To investigate whether this residue participates in an analogous protein-DNA interaction, an R26A RexT variant was made using site-directed mutagenesis (Supplementary Table 3). This R26A variant was then shown by EMSA and fluorescence anisotropy to have a significantly weaker affinity for DNA than wild-type RexT (Fig. 2e, Supplementary Fig. 5a, Supplementary Table 1–3). Gln56 from the N-terminal end of the α4 helix in NolR is involved in base-specific interactions with DNA and interacts with either an adenine or thymine base, facilitating the recognition of the NolR consensus motif (A/T)TTAG-N$_9$-A(T/A)[32]. Replacement of this residue with alanine in NolR causes a less than two-fold change in its affinity for DNA[32]. At the corresponding position in RexT is Lys50 (Fig. 2e, f). Change of this residue in RexT into alanine (K50A) also weakens the affinity of RexT for DNA (Fig. 2e, Supplementary Fig. 5a, and Supplementary Tables 1–3). Lys and Arg residues are known to predominantly interact with guanine bases, forming the most favorable interaction among the possible amino

acid–nucleotide interactions[34]. Correspondingly, RexT has been shown to bind a palindromic sequence <u>ATTCG</u>-N$_{15}$-<u>CGAAT</u>[17] and we hypothesize that Lys50 is important for DNA binding through an interaction with guanine. Finally, in NolR the main-chain amide of Gly46 on the α3 helix forms a hydrogen bond with the backbone phosphate of DNA[32]. Similarly, in both monomeric units of the reduced RexT structure, the mainchain amide of the corresponding Cys41 interacts with a molecule, which is best modeled as a chloride anion from the crystallization buffer (Supplementary Fig. 6). Here, the sidechain changes in the C40S and C41S RexT variants have only minor impacts on the ability of RexT to interact with DNA (Fig. 2e, Supplementary Fig. 5b, and Supplementary Tables 1–3).

**Identification of cysteine residues involved in sensing oxidative stress**. To identify the Cys residues that are involved in disulfide bond formation, the reactivity of three RexT Cys-to-Ser variants (C40S, C41S and C105S) and wild-type RexT towards H$_2$O$_2$ was probed using an assay to monitor consumption of H$_2$O$_2$ via the ferrous oxidation of xylenol orange (FOX assay) over time[9] (Fig. 3a–e, Supplementary Table 3). This assay has been previously described and used to show H$_2$O$_2$ consumption in OxyR[9]. Using similar conditions to those described for OxyR[9], it was determined that when 200 μM of H$_2$O$_2$ was mixed with 100 μM of either wild-type RexT or C105S RexT, approximately 50-percent of the added H$_2$O$_2$ was consumed in 200 s (Fig. 3b, c). In contrast, the C40S and C41S variants showed an impaired ability to consume H$_2$O$_2$ relative to wild-type RexT (Fig. 3d, e). These results along with our structural results, which show that Cys40 and Cys41 are located adjacent to each other on the α3 helix, suggest that oxidative stress results in disulfide bond formation between Cys40 and Cys41(Fig. 3a). The identification of the disulfide bond forming Cys residues was further explored using mass spectrometry (Supplementary Fig. 7). These experiments showed the formation of a disulfide bond in wild-type RexT and the C105S variant, but not in the C40S or C41S variants (Supplementary Fig. 7a, c–f).

Furthermore, the larger impairment of the C41S variant to consume H$_2$O$_2$, suggests it functions as the peroxidatic Cys residue, or the residue that is converted to a sulfenic acid moiety in the process of disulfide bond formation. This finding also suggests that Cys40 is the resolving Cys, or the residue that reacts with the formed sulfenic acid moiety to make a disulfide bond. This hypothesis was probed using mass spectrometry to look for incorporation of the small molecule dimedone into RexT during treatment with H$_2$O$_2$. Dimedone modifies sulfenic acid moieties and was shown to not be incorporated into the C105S variant (Supplementary Fig. 7b, f). In contrast, these experiments revealed that one equivalent of dimedone reacted with both wild-type RexT and the C41S variant (Supplementary Fig. 7c, e) and two equivalents of dimedone reacted with the C40S variant (Supplementary Fig. 7d). Collectively, the mass spectrometry experiments show that a sulfenic acid moiety can be formed at Cys41 and Cys105 in response to H$_2$O$_2$. The latter modification is likely formed due to the reactive nature of the non-disulfide bonded Cys residue in RexT. On the other hand, the C41 modification is observed only in the case where the proposed resolving Cys residue (Cys40) is absent (Supplementary Fig. 7d). This result is consistent with disulfide bond formation being faster than dimedone incorporation. The location of the Cys105 residue, which is only modeled in chain B of the determined structure is located ~20 Å away from both Cys40 and Cys41 (Fig. 3a). Based on this observation, the wild-type levels of H$_2$O$_2$ consumed by the C105S RexT variant, and a circular dichroism experiment that shows no large structural rearrangements

following the addition of H$_2$O$_2$ to wild-type RexT, we conclude that Cys105 is not involved in the RexT-mediated oxidative stress response (Fig. 3c and Supplementary Fig. 8). Intriguingly, despite the widespread use of Cys residues to coordinate metal ions in the ArsR-SmtB class of proteins, adjacent Cys residues are rarely used as metal-binding ligands[35]. Likewise, although there are cases in this protein family where adjacent Cys residues are used to coordinate methyl-As(III) or As(III), all of the involved cysteine residues are found at the end of α5 and the flexible region of the C-terminus[28,36]. These observations suggest that RexT has evolved as a redox sensor through formation of a specific sensory site for responding to H$_2$O$_2$.

**A vicinal disulfide-bond mediated conformational change in RexT**. To visualize the structural consequence of Cys residue oxidation, a 2.16-Å resolution structure of H$_2$O$_2$-treated RexT was solved (RexT_Ox, Table 1). These crystals were prepared by removing crystals of reduced RexT from the anaerobic chamber and immersing them in a solution of cryoprotectant that contained H$_2$O$_2$. The oxidized structure shares a similar overall architecture with that of reduced RexT (rmsd of 0.380 Å for 1258 atoms). In chain A of the RexT dimer, a slight orientation difference is observed for both the Cys40 and Cys41 sidechains relative to the reduced structure. In contrast, in the α3 helix region of chain B, in agreement with the above-described biochemical results, the electron density is consistent with the presence of a disulfide bond between Cys40 and Cys41 (Fig. 4a, Supplementary Fig. 9, Supplementary Fig. 10a–d). To form this disulfide bond, Cys41 undergoes a large conformational change: the sidechain flips −167.8° and moves 8.8-Å away from its position in the reduced structure (Fig. 4a). The observed disulfide bond belongs to a rare group of disulfide bonds known as vicinal disulfide bonds. Based on the succession of sidechain dihedral angles and the handedness of the sulfur–sulfur bond, this vicinal disulfide bond has a *Trans-Z* (Tz) geometry[35] (Fig. 4b). Formation of this vicinal disulfide bond results in the creation of an eight-membered ring with the peptide backbone and distortion of the peptide bond geometry. To bring the two sulfur atoms close together, the ω peptide bond in the Tz conformation twists −35° from the ideal 180° geometry. This distortion is larger than the average value of −17°[35] and similar to that observed (−29°) in 1,5-alpha-L-arabinanase[37] (PDB ID: 3CU9), the first protein for which the Tz conformation was documented (Fig. 4b). In addition to this observed peptide bond distortion, the movement of Cys41 disrupts the interactions seen in the reduced structure between the backbone and sidechain of Cys41 and residues His75 and Asn77 from the wing of the wHTH motif (Fig. 4c). The lack of a disulfide bond in chain A is proposed to be due to crystal packing, which would prohibit a comparable movement of these residues (Supplementary Fig. 9). Finally, both the oxidized and reduced structures of RexT have the highest B factors in the DNA binding features (Fig. 4d). This result may suggest that consistent with that observed for other wHTH motifs, this region is flexible in solution (Fig. 4d, Supplementary Fig. 10e, and Table 1).

**Identification of key residues involved in H$_2$O$_2$ binding and activation**. As described above, the formation of a disulfide bond by H$_2$O$_2$ typically starts with the oxidation of the peroxidatic Cys residue to a sulfenic acid. A subsequent step using a resolving Cys completes the process[9,38]. To visualize how H$_2$O$_2$ could access Cys40 and Cys41 in RexT to facilitate disulfide bond formation, the cavities present in the structures of RexT were calculated using PyMOL (Fig. 5a). Through this analysis, it was found that a cavity leads from the surface of the protein to a cluster of amino acids that includes Cys41 (Fig. 5a). This cavity contains one

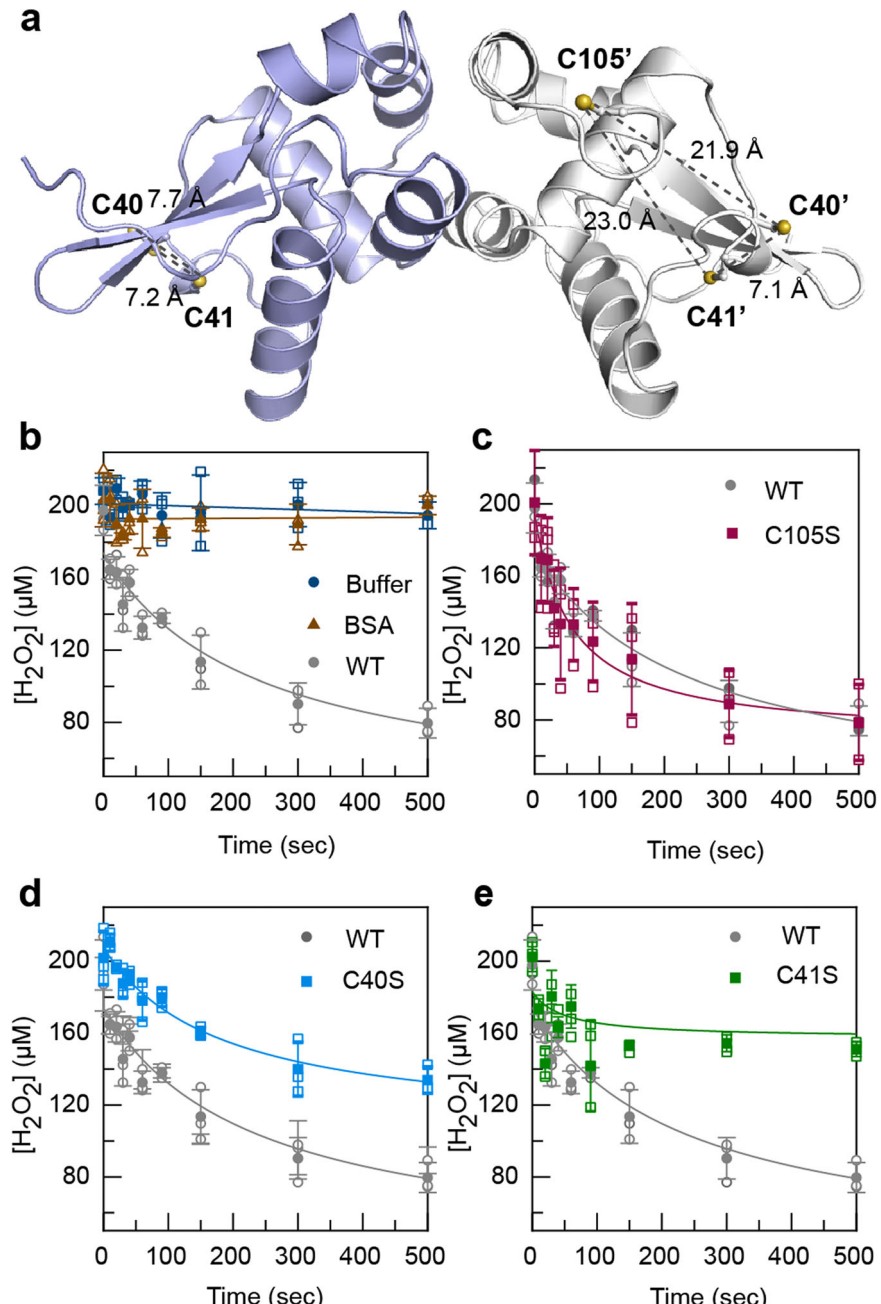

**Fig. 3 Cys40 and Cys41 form an intramolecular disulfide bond in response to H$_2$O$_2$ stress. a** The positions of the Cys residues in the reduced structure of RexT. The two sulfur atoms of Cys40 and Cys41 are located ~7 Å apart in both monomeric units of RexT. The sulfur atoms of Cys40 and Cys41 are located 21.9 Å and 23.0 Å away from the sulfur atom of Cys105, the last modeled residue in the crystal structure, respectively (shown for chain B of the structure). There are two measurements shown in chain A since the Cys41 residue sidechain shows two orientations of the sidechain. **b** Using the FOX assay, wild-type RexT was shown to consume H$_2$O$_2$ over time. Two control reactions are also included in this panel that shows H$_2$O$_2$ is not consumed in the absence of RexT or in the presence of the protein bovine serum albumin (BSA). **c** The C105S RexT variant consumes H$_2$O$_2$ similarly to wild-type RexT, suggesting it is not involved in mediating the oxidative stress response. **d** In contrast, the C40S variant of RexT shows a decreased ability to consume H$_2$O$_2$ relative to wild-type RexT. **e** As observed for the C40S variant, the C41S RexT variant is also impaired in its ability to consume H$_2$O$_2$, albeit to a greater extent. In **b**–**e**, data was measured using $n = 3$ independent experiments and is presented with the individual measurements (open shapes) and as the mean value of these measurements ± SD (closed shapes). Source data are provided as a Source Data file.

molecule of H$_2$O$_2$ in chain A of the H$_2$O$_2$-treated structure and one molecule of glycerol in chain B of the reduced structure, suggesting the possibility that H$_2$O$_2$ gains access to Cys41 using this route (Fig. 5a, b and Supplementary Fig. 11). Importantly, along this calculated cavity, a few polar and charged residues are arranged in a way that is reminiscent of the catalytic residues found in *Corynebacterium glutamicum* OxyR[9] (*Cg*OxyR, Fig. 5c).

For example, in RexT, the positioning of Arg61 and Thr68 resembles the conserved catalytic residues Arg278 and Thr107 in *Cg*OxyR that contribute key hydrogen bonds for the binding of H$_2$O$_2$ (Fig. 5b, c). The sidechain of Asn77 is close to Cys41, suggesting that like Gln210 in *Cg*OxyR, it could be involved in stabilizing a negatively charged Cys41 thiolate. Additionally, there is potential for His75 to act as a proton acceptor, either for the

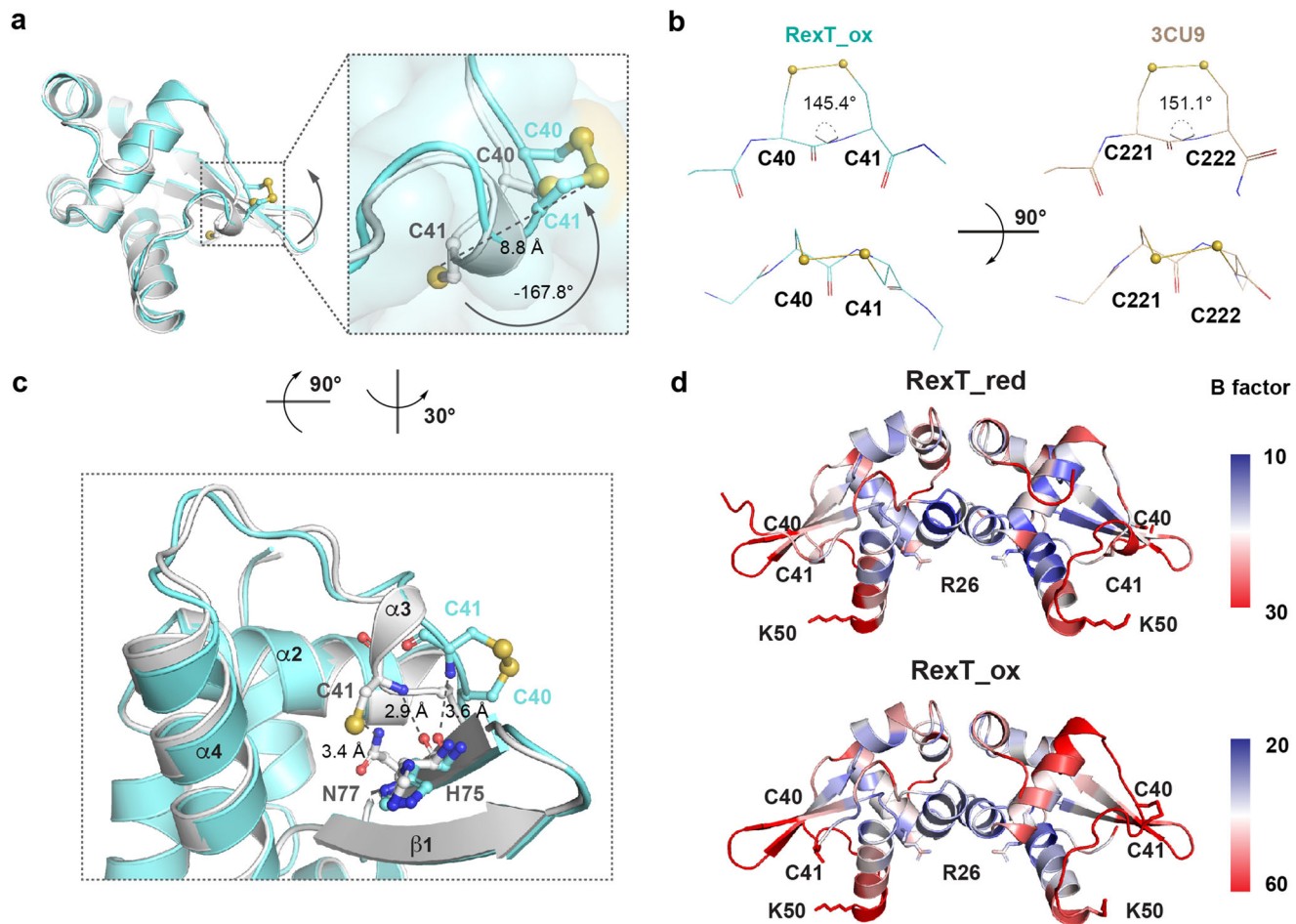

**Fig. 4 Disulfide bond formation in RexT causes a conformational change and loss of interactions. a** An overlay of chain B from the structures of the reduced (light gray) and oxidized (teal) states of RexT. A box is included to highlight the Cys residues involved in disulfide bond formation. The formation of the disulfide bond requires Cys41 to undergo a large conformational change. **b** Comparison of the eight-membered vicinal disulfide bond geometry in RexT and 1, 5-alpha-L-arabinanase[37] (PDB ID: 3CU9). The ω angles of both RexT and 1, 5-alpha-L-arabinanase show a large deviation from the ideal values. **c** Upon disulfide bond formation in RexT, the movement of Cys41 causes disruptions in the interactions between the thiol-group and the sidechain of Asn77 as well as interactions between the Cys41 backbone amide and the backbone of His75. Both Asn77 and His75 are found in the wing portion of the wHTH motif involved in DNA binding. **d** The B factors of the reduced (top panel, RexT_red) and oxidized (bottom panel, RexT_ox) states of RexT are illustrated in a continuum from blue, white to red. Blue corresponds to a lower B factor and red represents a higher B factor. The reduced structure is colored on a scale from 10 to 30, whereas the oxidized structure is colored on a scale from 20 to 60.

Cys41 thiol group or for $H_2O_2$ (Fig. 5b, c). Likewise, Gln74, although located outside of the cavity, is found in the vicinity of Cys40 and may also be involved in disulfide bond formation (Fig. 5b and Supplementary Fig. 11). To test the involvement of these residues in $H_2O_2$-mediated disulfide bond formation, five variants of RexT (R61A, T68A, N77A, Q74A, and H75A) were created and subjected to the above-described $H_2O_2$ consumption assay (Fig. 5d–h and Supplementary Table 3). The results of this experiment show that Arg61, Thr68, and Asn77 are important players in permitting RexT to consume $H_2O_2$ (Fig. 5d–f). Gln74 and His75, on the other hand, have little effect on the peroxidatic reactivity of RexT (Fig. 5g, h). Thus, remarkably, unlike other members of the ArsR-SmtB family of regulators, RexT has evolved to contain a Cys-Arg-Thr-Asn tetrad of residues to activate and reduce $H_2O_2$ and facilitate the formation of a sulfenic acid moiety.

**RexT represents a distinct class of regulatory redox sensors in the ArsR-SmtB family**. To investigate the prevalence of the RexT-based mode of gene regulation in the ArsR-SmtB family, we constructed a sequence similarity network (SSN)[39] based on the conserved HTH ArsR-type DNA-binding domain (IPR001845). This network mainly comprises two Pfam groups to which most of the previously characterized ArsR-SmtB regulators belong (Fig. 6, Supplementary Fig. 12, and Supplementary Table 4). At a cut-off value of $e^{-20}$, a main cluster of nodes can be visualized that contains most of the canonical metal- and arsenite-binding regulators. As described above for SeSmtB, regulators that contain Asp/Glu/His-rich metal-binding motifs on the α5 helix are found in two connected subgroups (Fig. 6, circle $I_a$-$I_b$, Supplementary Fig. 13). The first subgroup (circle $I_a$) includes canonical metal-binding regulators SmtB[19,31], CzrA[31], and NmtR[40] that contain a $DXHX_{10}HXXE/H$ metal-binding motif on the α5 helix (α5 site). Regulators that contain the α5 site and an additional Cys-rich metal-binding motif that bridges the N-terminus of one subunit with the α3 helix of a second subunit (α3N site), including CadC[27], ZiaR[41], AztR[42], and BxmR[43], are also found in this subgroup. The second subgroup (circle $I_b$) is less well characterized, and contains three regulators from *Mycobacterium tuberculosis*, one of which (*Mt*KmtR[44]), is known to use an $HX_6DHX_5EX_6HH$ metal-binding motif on α5. Aside from these

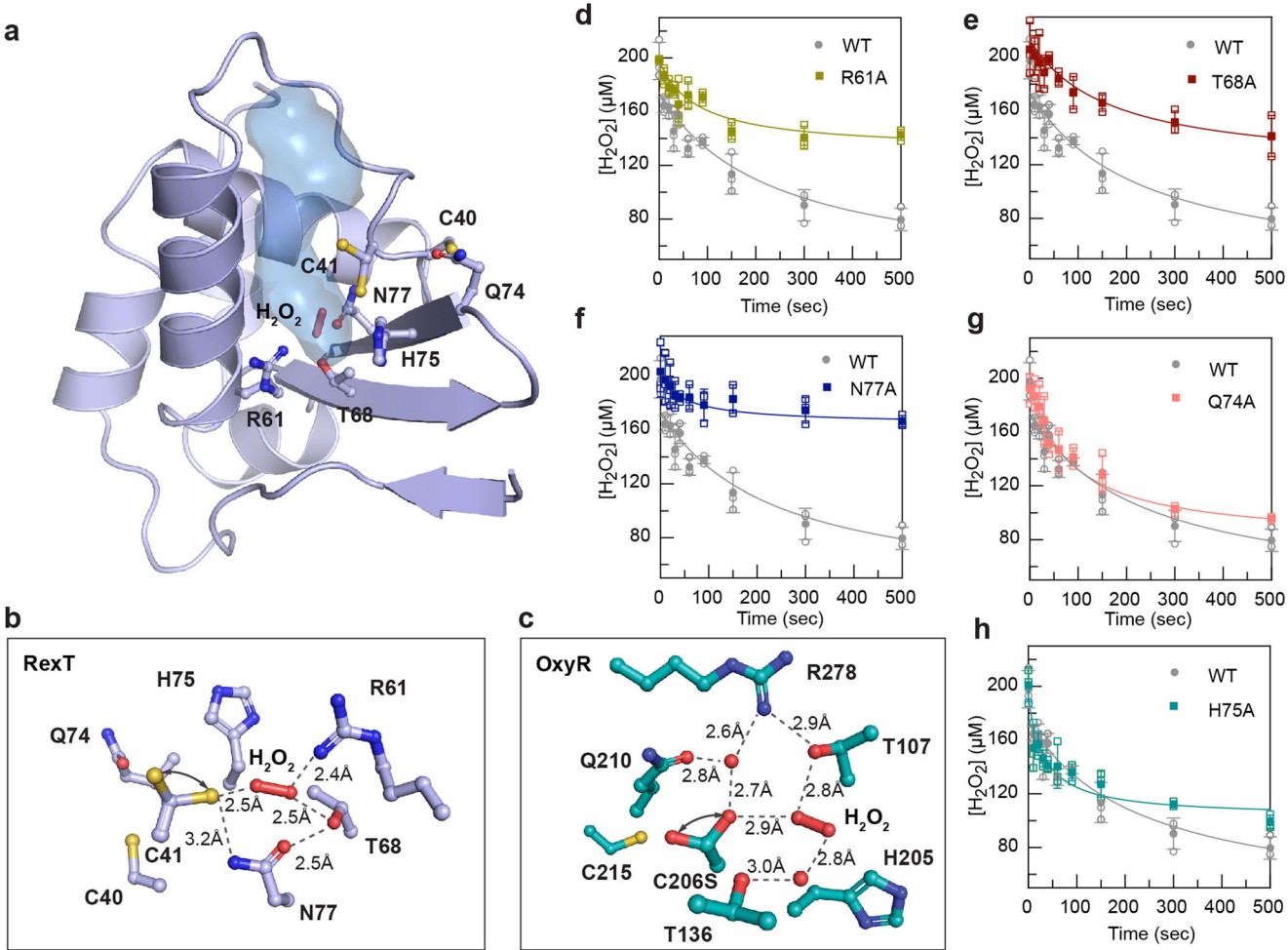

**Fig. 5 The structure of RexT reveals key residues involved in the H₂O₂-based response. a** A surface cavity (blue) is identified that leads from the surface of RexT to the Cys 41 residue. The calculated cavity in chain A contains a molecule of H₂O₂. The same channel, in chain B of reduced RexT, contains a molecule of glycerol from the cryoprotectant (*see* Supplementary Fig. 11a–d). **b** The bottom of the calculated cavity that surrounds Cys41 contains residues arranged in a way that suggests they are important players in the activation of H₂O₂. Arrows indicate two modeled conformations of Cys41. Electron density maps for this panel can be found in Supplementary Fig. 11e, f. **c** Similar to the key catalytically relevant residues observed in RexT, catalytically relevant Arg278 and Thr107 residues are found in OxyR[9] (PDB: 6G4R). Arrows indicate two modeled conformations of Ser206. **d–f** The R61A, T68A, and N77A RexT variants show changes in their ability to consume H₂O₂ relative to wild-type RexT, suggesting they are involved in mediating the oxidative stress response. **g, h** Both the Q74A and H75A RexT variants consume H₂O₂ similarly to wild-type RexT, suggesting they are not key players in activating and reducing H₂O₂. The control reactions shown in Fig. 3b are also relevant to **d–h** in this figure. In **d–h**, data was measured using $n = 3$ independent experiments and is presented with the individual measurements (open shapes) and as the mean value of these measurements ± SD (closed shapes). Source data are provided as a Source Data file.

regulators, the main cluster of nodes contains a second region rich in regulators that use two or three Cys residues to bind an arsenite or a methyl-arsenite anion (Fig. 6, circle II). These Cys residues are found in different locations, including the α3 helices of *Ec*ArsR[18] and the arsenite-binding regulator AseR from *Bacillus subtilis* (*Bs*AseR[45]). They are also found on the N-terminus and the loop connecting the α2 and α3 helices of the ArsR homolog from *Corynebacterium glutamicum* (*Cg*ArsR[28]), or on the C-terminal extension of *Af*ArsR[28] and the ArsR homolog from *Shewanella putrefaciens* (*Sp*ArsR[36]) (Supplementary Fig. 13). Each of these regulators highlights the flexibility and versatility of the wHTH-ArsR architecture (Supplementary Fig. 13, Supplementary Table 4). Finally, the main cluster of nodes also contains an emerging class of regulators that do not bind a metal or arsenite ion at all, further emphasizing the adaptability of the protein fold (Fig. 6, circle III). In this group, BigR[46] and SqrR[47] are implicated in responding to the presence of hydrogen sulfide (H₂S) and reactive sulfur species (RSS) using

two-Cys residues from the α2 helix and the α5/C-terminal extension. This pair of Cys residues are also conserved in HlyU[29,30] and YgaV[48], which are found in a similar region of the main cluster and have been suggested to be involved in sensing ROS using an unknown regulatory mechanism (Supplementary Fig. 13). Interestingly, NolR[32], which is highlighted in Supplementary Fig. 4, is a global regulator for symbiotic nodulation and is also found close to these nonmetal-binding regulators. However, NolR does not contain any Cys residues and the environmental signal that triggers gene regulation is unknown (Supplementary Fig. 13, Supplementary Table 4)[32].

Outside of this main cluster, a multitude of additional clusters contain at least one biochemically or structurally characterized protein from this superfamily. For example, CmtR[49], which uses an α4 and N-terminal metal-binding site, is found on a minor cluster separated from the main body of the network (Fig. 6 and Supplementary Fig. 13). In this work, it was found that despite playing a role in the ROS response like that suggested for YgaV

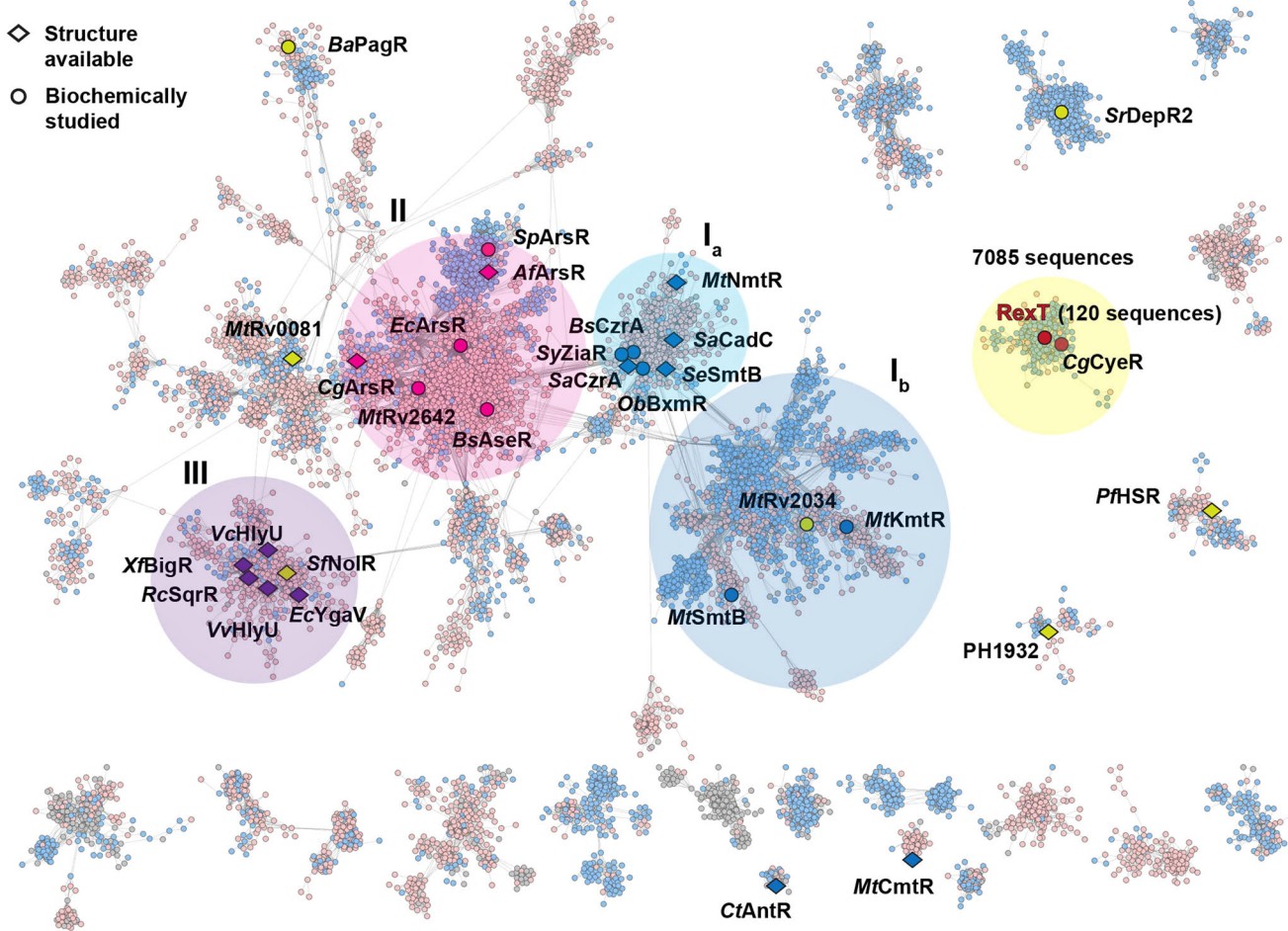

**Fig. 6 A sequence similarity network (SSN) of the ArsR-SmtB regulator family reveals RexT forms a distinct class of its own.** The SSN[39] was constructed using the EFI server[81,85,86] with 20616 UniRef50 sequences that contain an annotated "HTH ArsR-type DNA-binding domain" (IPR001845) and analyzed by Cytoscape[83] with a cut-off value of e$^{-20}$. The nodes are colored by their Pfam classification: pink nodes represent the 10600 sequences from PF01022 (HTH_5) and blue nodes represent the 7564 sequences from PF12084 (HTH_20). Only the main body of the network that contains proteins of interest are shown. Regulators that have been previously biochemically or structurally characterized are highlighted in circle and diamond shapes, respectively. These shapes are color-coded to denote the signal that these regulators sense (blue: metal-binding; pink: arsenite-binding; purple: putatively redox-based; yellow: unknown mechanisms) The main cluster of nodes is colored by large circles to differentiate regulators that use the α5 helix to bind metal ions (circles I$_a$ and I$_b$), Cys residues to bind an arsenite or a methyl-arsenite anion (circle II), or do not bind a metal ion at all (circle III). Organism abbreviations: Af *Acidithiobacillus ferrooxidans*, Ba *Bacillus anthracis*, Bs *Bacillus subtilis*, Cg *Corynebacterium glutamicum*, Ct *Comamonas testosterone*, Ec *Escherichia coli*, Mt *Mycobacterium tuberculosis*, No *Nostoc* sp. PCC 7102, Ob *Oscillatoria Brevis*, Pf *Pyrococcus furiosus*, PH *Pyrococcus horikoshii*, Rc *Rhodobacter capsulatus*, Rf *Rhizobium fredii*, Sy *Synechocystis* sp. PCC 6803, Se *Synechococcus elongatus* PCC 7942, Sa *Staphylococcus aureus*, Sp *Shewanella putrefaciens*, Sr *Streptomyces filamentosus*, Vc *Vibrio cholerae*, Vv *Vibrio vulnificus* CMCP6, Xf *Xylella fastidiosa*. See Supplementary Fig. 12, 13, and Supplementary Table 4 for additional information.

and HlyU, RexT is separated from the main cluster of nodes, representing a distinct class of its own. The RexT cluster in the SSN contains 7085 non-redundant protein sequences in the UniProtKB database (Fig. 6). To date, RexT is the only member of this cluster that has been biochemically and structurally investigated. However, as it is known that the formation of a disulfide bond under oxidative stress in RexT induces the expression of a thioredoxin-encoding gene, it was hypothesized that the co-occurrence of *rexT* and *trx* genes is a genomic signature of the disulfide bond-based redox sensor (Supplementary Fig. 14, Supplementary Table 4). Analysis of the genomic neighborhood diagram of the 7085 sequences that belong to the RexT cluster revealed that 105 sequences are likely to behave similarly to RexT and act as a redox regulator that responds to ROS (Supplementary Fig. 14, 15a, Supplementary Table 5). Remarkably, all 105 sequences contain at least two-Cys residues: Cys41 is found in all 105 putative RexT proteins whereas Cys40 is

found in only 85 of the sequences (based on RexT numbering, Supplementary Fig. 15a, b, Supplementary Table 6). Of the 105 sequences that contain Cys41, two are from unclassified bacteria, one is from Abditibacteria, and 102 are from Cyanobacteria, suggesting the disulfide bond-based redox switch is unique to Cyanobacteria (Supplementary Fig. 14, Supplementary Table 5). Further analysis shows that the 102 sequences mainly fall into three orders of Cyanobacteria: 84 sequences come from *Nostocales*, nine sequences come from *Synechoccocales*, three sequences come from *Oscillatoriales*, and the remaining six sequences are from unclassified Cyanobacteria species. The dual Cys40-Cys41 sequence signature on the α3 helix is conserved in 83 of the 84 *Nostocales* sequences (Supplementary Fig. 15b). In contrast, the dual-Cys motif is not observed in *Synechoccocales* or *Oscillatoriales* (Supplementary Fig. 15c). In the latter two orders, conserved Cys residues are found either at positions 41 and 68 (8 out of 11), positions 9 and 41

(5 out of 11), or at positions 41, 44, and 105 (Supplementary Fig. 15c, Supplementary Table 5).

Consistent with these sequence-specific observations, a rooted phylogenetic tree of RexT homologs shows that sequences of *Nostocales* form different clades than the *Synechoccocales* or *Oscillatoriales* order sequences (Supplementary Fig. 16). Interestingly, the one *Nostocales* sequence that does not contain the Cys40-Cys41 motif (but instead contains C40 and C105) lies at the branching point between the *Nostocales* and the *Synechoccocales/Oscillatoriales* sequences, suggesting an evolutionary diverging point between these orders (Supplementary Fig. 16).

## Discussion

Here, we used biochemical, structural, and bioinformatics studies to elucidate the molecular details of how RexT senses $H_2O_2$ and initiates an oxidative stress response. It was found that RexT senses $H_2O_2$ through the formation of a vicinal disulfide bond and the molecular details of the disulfide bond-based redox regulation process were revealed. Importantly, RexT differs from members of the ArsR-SmtB family of regulators that have been thus far characterized and represents the only sensor in this family that has been shown to employ catalytic residues to facilitate a response to ROS. Thus, this work adds to the regulatory repertoire of the ArsR-SmtB metalloregulator superfamily.

Mechanistically speaking, most $H_2O_2$-mediated disulfide bond-forming reactions proceed through an attack of the deprotonated-negatively charged Cys sidechain thiolate on $H_2O_2$ to form an unstable sulfenic acid moiety (Fig. 7)[9,50]. This transient species can subsequently react with a nearby Cys residue to form an intramolecular disulfide bond (Fig. 7). For RexT, based on the mutagenesis studies, mass spectrometry experiments, and the

**Fig. 7 A mechanistic proposal for $H_2O_2$-mediated disulfide bond formation in RexT.** Based on the crystal structures of oxidized and reduced RexT, as well as the $H_2O_2$ consumption assays performed in this work, we propose that a disulfide bond is formed between Cys40 and Cys41. In addition, the first step of disulfide bond formation, or the conversion of the Cys41 sidechain into a sulfenic acid moiety requires a tetrad of residues, Cys41, Arg61, Thr68, and Asn77.

X-ray crystal structures determined in this work, we posit that Cys41 is the peroxidatic Cys and that Asn77, due to it being within hydrogen-bonding distance of Cys41, is involved in forming or stabilizing the negatively charged Cys41 thiolate (Fig. 5b and Fig. 7). Arg61 and Thr68, on the other hand, are suggested to form hydrogen bonds with $H_2O_2$ and favor its reduction by polarizing the O–O bond and facilitating the requisite $S_N2$ reaction (Fig. 5b and Fig. 7)[50]. Consistent with the importance of these residues in disulfide bond formation, Asn77 and Arg61 are completely conserved in all identified RexT homologs that are adjacent to a thioredoxin gene (Supplementary Fig. 15a). Thr68, although not completely conserved, is replaced by a Ser residue in some sequences from the *Nostocales* order or intriguingly substituted by a Cys residue in some RexT homologs from *Synechococcales* and *Oscillatoriales* that lack Cys40 (Supplementary Fig. 15b, c).

As described above, our mechanistic proposal for RexT is reminiscent of that put forward for OxyR[9], which has been suggested to also rely on Thr and Arg residues to activate $H_2O_2$ for reduction (Fig. 5b, c). However, unlike OxyR, the rate of $H_2O_2$ consumption by RexT appears to be slower[9]. Specifically, it was observed that it took wild-type RexT about 200 s to consume half of the added $H_2O_2$ in our assays (Fig. 3 and Fig. 5). For OxyR, using the same assay, an equimolar amount of $H_2O_2$ is consumed within 10 sec[9]. The magnitude of difference between the rates of $H_2O_2$ consumption in these proteins is proposed to correlate with the different levels of cellular $H_2O_2$ in the organisms that use these regulators. For example, OxyR is mainly found in Gram-negative bacteria where the $H_2O_2$ levels are maintained at a low level by the efficient antioxidant activity of catalase. In *E. coli*, this statement means that OxyR is activated when exogenous $H_2O_2$ levels accumulate to ~3 μM or intracellular levels near a concentration of 0.2 μM[7,51,52]. In contrast, it has been shown that the RexT-mediated expression of thioredoxin occurs at $H_2O_2$ concentrations above 1 mM[17]. This higher concentration can presumably be attributed to studies that show *Anabaena* (from the order *Nostocales*) predominantly uses two-Cys peroxiredoxin proteins to cope with detoxification of $H_2O_2$ rather than catalase[53]. These peroxiredoxin proteins showcase modest catalytic rates and are sensitive to overoxidation, which allows higher concentrations of $H_2O_2$ to accumulate[53]. Similarly, as it is currently proposed that the RexT-TrxA2 system could function in both vegetative cells and heterocysts[17], it is worth noting that in heterocysts, detoxification of $H_2O_2$ involves a rubrerythrin that showcases peroxidase activity. Recent work on this rubrerythrin revealed a measured $K_M$ of 2 mM $H_2O_2$[54]. Thus, regardless of whether RexT functions in vegetative cells or in heterocysts, it appears optimally tuned to start the signaling process at higher $H_2O_2$ levels than OxyR. At the molecular level, it is also possible that the lower observed rate of $H_2O_2$ consumption in RexT is due to the different positioning of $H_2O_2$ in the active site relative to that seen in OxyR. In OxyR, the oxygen atom of $H_2O_2$ that is attacked by the nucleophilic Cys residue engages in multiple hydrogen-bonding interactions with active site residues (Fig. 5c)[9]. These interactions presumably increase the electrophilicity of the oxygen atom and thereby facilitate the $S_N2$ displacement reaction. The equivalent oxygen atom of $H_2O_2$ observed in RexT does not engage in any such interactions (Fig. 5b).

Last, it was determined that Cys41 is the peroxidatic Cys residue, whereas Cys40 is the resolving Cys residue (Figs. 3, 5, and 7). Consistently, we find that Cys41 is absolutely conserved among all "true" RexT homologs or those that are adjacent to a thioredoxin-encoding gene (Supplementary Fig. 15). In contrast, despite conservation of most of the identified residues involved in dimer formation, DNA binding (Arg26 and Lys50), and the required residues for $H_2O_2$ reduction (Asn77 and Arg61), Cys40 is not

conserved in all these identified homologs (Supplementary Fig. 15). Intriguingly, CyeR[55], which is also found in the RexT cluster of the SSN, upregulates the expression of the old-yellow-enzyme-family protein-encoding gene *cye1* in response to oxidative stress (Fig. 6 and Supplementary Fig. 17). This CyeR-based mode of regulation is native to *Mycobacterium* and *Streptomyces* and its inactivation is known to rely on one Cys residue that is found in an equivalent position to Cys41 in RexT (Supplementary Fig. 17). This protein has been suggested to function as a monomer and form an intramolecular disulfide bond with a second Cys that is found seven residues away, indicating the possibility that some RexT homologs could function through the formation of an alternate disulfide linkage[55] (Supplementary Fig. 15 and 17). The conservation of just one Cys residue among RexT homologs also parallels what is known about OxyR, for which there exist both two-Cys and one-Cys containing homologs[56,57]. As with the CyeR-mediated response, the molecular details of how these one-Cys OxyR homologs sense and respond to $H_2O_2$ remain enigmatic.

In this work, we also found that the dimeric architecture and DNA binding regions of RexT are relatively unchanged between the reduced and oxidized states (Fig. 2c and Supplementary Fig. 18a). This result is unlike the conformational changes observed in OxyR, where formation of a disulfide bond causes a global conformational change in the protein architecture[9] (Supplementary Fig. 18e, f). Instead, in RexT, the formation of an eight-membered vicinal disulfide bond results in only a local reorganization of the α3 helix and disruption of its interactions with the "wing" of the wHTH motif (Fig. 4). The lost interactions between the α3 helix and β2 strand in the wHTH motif in the oxidized structure, suggest that Cys41 is a redox switch: its movement propagates as a series of small movements in the α3 helix that disrupt interactions with the other structural elements required for DNA binding (Fig. 4 and Supplementary Fig. 10e). In line with this statement, a vicinal disulfide bond has also been observed in redox-cycling proteins as well as in proteins where the rearrangement to form the strained eight-membered ring constitutes the required conformational change for protein inactivation[35]. Collectively, these changes should disfavor DNA binding, and initiate the transcription of *trxA2*. We acknowledge that the magnitude of the observed conformational change may be constrained by the crystal lattice, but the absence of a large conformational change upon signal recognition is not uncommon for ArsR-SmtB family[58] (Supplementary Fig. 8 and Supplementary Fig. 18b–d). For example, in the Zn-binding CzrA and SmtB regulators, small changes in the internal motions have been linked to the formation of a protein conformation that has a lower affinity for DNA[33,59] (Supplementary Fig. 18b, d). Likewise, in SqrR, it has been proposed that reception of the environmental signal results in small structural perturbations that energetically disfavor DNA binding[47] (Supplementary Fig. 18c). Finally, it is also important to point out that formation of a disulfide bond in RexT has other implications. It has been previously shown that TrxA2 is the native reducing agent for oxidized RexT[17]. In the oxidized structure, the newly formed disulfide bond is oriented towards the protein surface and perhaps coupled with the architectural changes in RexT, more favorably interacts with reduced TrxA2 and permits subsequent resolving of the disulfide linkage (Fig. 4a).

With regards to the third Cys residue found in RexT (Cys105), our bioinformatics analysis revealed that only 38 out of 85 *Nostocales* sequences have this residue, and its occurrence seems to be randomly distributed among different families under the *Nostocales* order (Supplementary Fig. 16). Interestingly, a multiple sequence alignment between different types of ArsR-SmtB regulators shows that this C-terminal Cys residue is conserved in a

few cases. For example, the equivalent of Cys105 is found as part of the arsenite-binding motif of *Af*ArsR (Supplementary Fig. 13). Likewise, although not perfectly aligned, a C-terminal Cys residue of CmtR is known to be involved metal ion binding (Supplementary Fig. 13). A C-terminal Cys residue has also been shown in BigR and SqrR to be involved in binding a reactive sulfur species in partnership with another Cys residue on the α2 helix (Supplementary Fig. 13). There are also several examples (*Ec*ArsR[18] and *Bs*AseR[45]), in line with what we hypothesized for RexT, which don't seem to have a function for this C-terminal Cys residue (Supplementary Fig. 13). Assuming multiple ArsR-type paralogs within an organism arise from gene duplication during evolution (like *M. tuberculosis*, which contains 12 putative ArsR-SmtB regulators[60,61]), it seems likely that the Cys105 residue is present in an ancestral ArsR sequence and is an evolutionary remnant in RexT homologs. On the other hand, the phylogenetic tree suggests that the Cys40 residue may be a gain of function for the *Nostocales* organisms, which unlike *Synechococcoles* and *Oscillatoriales*, use heterocysts for nitrogen fixation and require thioredoxin to maintain an anaerobic environment (Supplementary Fig. 16).

The continuous recycling, repurposing, and adaptation of a common protein fold to a new function has been suggested to account for how a proposed group of <1000 protein folds can give rise to the vast number of proteins found in living organisms[62,63]. This concept is embodied by the ArsR-SmtB family of regulators, which allow organisms to adapt to environmental conditions and respond to environmental stressors. To date, members of the ArsR-SmtB family of regulators family, have been shown to exploit a common protein scaffold to sense and respond to environmental stimuli, including the availability of metal ions, arsenite ions, and RSS (Fig. 6). In this work, we describe how Nature has yet again exploited the ArsR-SmtB regulator architecture to sense, activate, and reduce $H_2O_2$ to begin the oxidative stress response in cyanobacteria (Fig. 5). In this case, it was found that placement of two-Cys residues adjacent to one another, an arrangement that is typically not conducive to metal-binding[35], allows for formation of a vicinal disulfide bond in response to increased levels of $H_2O_2$. This result, the identification of RexT homologs that may rely on a different set of Cys residues, and the vast uncharted territory in the ArsR-SmtB SSN (Fig. 6), makes it clear that we will continue to be surprised by examples of how the ArsR-SmtB fold can be reused, repurposed, and reinvented.

## Methods

**Protein expression and purification for RexT and selenomethionine-labeled RexT (SeMet-RexT).** The *Nostoc* (*Anabaena*) sp. PCC 7120 rexT gene (Locus: alr1867, GenBank: BA000019.2) was codon optimized for *E. coli* expression, commercially synthesized, and sub-cloned into pET21d(+) vector with NcoI and XhoI restriction sites by GenScript. The resulting construct of RexT contains an N-terminal 6x-His-tag and a TEV protease cleavage site (ENLYFQG). *E. coli* BL21(DE3) cells were transformed with the plasmid carrying the rexT gene and selected on LB agar plates containing 50 µg/mL ampicillin. A single colony was inoculated into a 5 mL LB starter culture and grown at 37 °C for 16 h. The 5 mL starter culture was subsequently diluted into 1 L of LB medium that contained 50 µg/mL ampicillin and grown at 37 °C with a shaking rate of 200 rpm until the $OD_{600}$ reached 0.6. At this point, the 1 L culture was cooled to 16 °C and isopropyl β-D-1-thiogalactopyranoside (IPTG) was added to the culture at a final concentration of 100 µM to induce protein expression. After 16 h of incubation at 16 °C with a shaking rate of 200 rpm, the cells were harvested by centrifugation (5000 × *g*, 20 min) at 4 °C. 2.5 g of wet cell pellet was typically obtained from a 1 L growth.

To purify His-tagged RexT (His-RexT), the cell pellets (~5 g) were resuspended in 50 mL of Buffer A (20 mM tris(hydroxymethyl)aminomethane (Tris, pH 8.0), 1 M NaCl, 25 mM imidazole, 10 mM β-mercaptoethanol (BME), and 5% v/v glycerol) which was supplemented with 1 mM phenylmethylsulfonyl fluoride (PMSF). The resuspended cells were sonicated on ice with a 3 s pulse followed by 15 s rest for a total pulse time of 6 min at 30% output (Fisherbrand™ Model 120 Sonic Dismembrator). The cell debris was removed by centrifugation (12,000 × *g*, 20 min) twice. The supernatant was loaded on a gravity Nickel column (Ni

Sepharose 6 Fast Flow resin, Cytiva) with a bed volume of 10 mL, which was pre-equilibrated with Buffer A. After the supernatant passed through the column, the column was washed three times with 10 mL of Buffer A. His-RexT was eluted with 30 mL of Buffer B (20 mM Tris (pH 8.0), 1 M NaCl, 200 mM imidazole, 10 mM BME, and 5% v/v glycerol). Ethylenediaminetetraacetic acid (EDTA, pH 8.0) and dithiothreitol (DTT) were added to the eluate to a final concentration of 0.5 mM and 2 mM, respectively. To remove imidazole, the eluate was concentrated by centrifugal filters (10 kDa, MilliporeSigma) at $5000 \times g$ and buffer-exchanged with an Econo-Pac 10DG column (Bio-Rad) pre-equilibrated with Buffer C (50 mM Tris (pH 8.0), 100 mM NaCl, 5% v/v glycerol, and 2 mM DTT). To cleave the His-tag, His-RexT, and TEV protease (purified based on a published protocol[64]) were added to TEV digestion buffer (50 mM Tris (pH 8.0), 0.5 mM EDTA, and 2 mM DTT) to a final volume of 50 mL. The resultant mixture was incubated on a table-top shaker at 4 °C for 16–20 h. To remove TEV protease after the digestion was complete, the TEV digestion mixture was loaded onto an anion exchange column (Enrich™ Q, Bio-Rad) pre-equilibrated with Buffer D (20 mM 4-(2-hydroxyethyl)-1-piperazine-ethanesulfonic acid (HEPES, pH 8.0), 5% v/v glycerol, and 2 mM DTT). The flow-through fraction was collected, and the column was washed with Buffer D and eluted with a 0–100% gradient of Buffer E (20 mM HEPES (pH 8.0), 1 M NaCl, 5% v/v glycerol, and 2 mM DTT). TEV was eluted at a concentration above 40% Buffer E, whereas tag-free RexT (RexT) existed mainly in the flow-through fraction and to a lesser extent in the lower Buffer E gradient fractions. Fractions of RexT were pooled, concentrated and loaded onto a size-exclusion column (HiLoad™ 16/600 Superdex 200 pg column, Cytiva) pre-equilibrated with Buffer F (50 mM HEPES (pH 7.5), 100 mM NaCl, 5% v/v glycerol, and 2 mM DTT). The dimeric fraction of RexT eluted from the size-exclusion column was concentrated, flash-frozen in liquid $N_2$, and stored in a −80 °C freezer. RexT used for crystallographic studies was further buffer-exchanged into Buffer G (20 mM HEPES (pH 7.5) and 5 mM DTT) with an Econo-Pac 10DG column. Protein purity was checked using a 16% polyacrylamide Tris-Tricine Gel and estimated to be more than 95% pure. The concentration of RexT was measured using the absorbance at 280 nm and the extinction coefficient of the fully reduced state ($2980\ M^{-1}\ cm^{-1}$) calculated by ProtParam[65].

The conditions for overexpression of selenomethionine-labeled RexT (SeMet-RexT) are similar to those of RexT, except that a methionine auxotrophic E. coli B834 (DE3) cell line was used. Consequently, a growth medium which consisted of SelenoMet™ medium base and nutrient mix (Molecular Dimensions) was used in place of LB, with 40 mg/L L-methionine added before IPTG induction. Once the $OD_{600}$ reached 0.6, the growth medium was removed by centrifugation and replaced with the same medium supplemented with 40 mg/L L-selenomethionine (TCI). The purification procedure of SeMet-RexT is identical to that of RexT.

**Crystallization and X-ray structure determination of SeMet-RexT, RexT, and oxidized RexT (RexT_ox).** A mosquito pipetting robot (TTP LabTech) was used to screen crystallization conditions anaerobically for RexT. First RexT crystals were observed following anaerobic incubation in a 20 °C incubator located in a Coy chamber ($O_2 < 20$ ppm, Coy Lab) at 20 °C for a week. This condition contained 0.3 uL of 20 mg/mL RexT and 0.3 uL of crystallization solution (0.2 M NH₄Cl (pH 6.3) and 20% v/v PEG3350). In subsequent optimization, 1 μL of a modified crystallization solution (10 mM 2-(N-morpholino) ethane sulfonic acid (MES, pH 6.3), 0.2 M NH₄Cl, 20% (v/v) PEG3350 and 5% v/v glycerol) was mixed with 1 μL of 20 mg/mL RexT using sitting drop vapor diffusion. RexT crystals formed within a week of being stored in a 20 °C incubator located in the anaerobic Coy chamber. These crystals were cryo-protected with a solution of 10 mM MES (pH 6.3), 0.2 M NH₄Cl, 20% PEG3350, and 30% v/v glycerol and quickly looped and frozen using liquid $N_2$ in the Coy chamber. SeMet-RexT crystals were produced and harvested similarly. To obtain "oxidized" RexT (RexT_ox) crystals, the crystallization trays were taken out of the Coy chamber and the crystals were looped with an $H_2O_2$-containing cryoprotectant solution (10 mM MES (pH 6.3), 0.2 M NH₄Cl, 20% PEG3350, and 24–32% v/v glycerol with 1.6, 16, 160, 1600, or 2000 mM $H_2O_2$) within 2 h. If the crystal trays were stored outside of the Coy chamber for >1 day, the crystals would diffract in patterns that differ from those looped inside of the Coy chamber and could not be processed, indicating potential oxidative damage on these crystals.

The datasets for SeMet-RexT. RexT, and RexT_ox were collected at the Life Sciences Collaborative Access Team beamlines 21-ID-F (Rayonix MX-300) and 21-ID-G (Rayonix MX-300) at the Advanced Photon Source, Argonne National Laboratory. All datasets were collected at a temperature of 100 K and wavelengths of 0.97872 Å (RexT), 0.97857 Å (SeMet-RexT) and 0.97857 Å (RexT_ox).

A 2.50-Å resolution dataset of SeMet-RexT was indexed, integrated, and scaled in XDS[66,67]. This initial dataset was indexed as $P6_4$ and according to Xtriage[68] showed 92.8% anomalous completeness as well as a good anomalous signal to 3.0-Å resolution. This resolution was input into Phenix Hybrid Substructure Search (HysS)[69,70] and the positions of three Selenium atoms distributed between the two RexT protomers present in the asymmetric unit were identified. The produced heavy-atom site file was input into Phenix AutoSol and used to generate experimental maps that had an overall figure-of-merit (FOM) of 0.363 for the full resolution range of the data[71–73]. These maps were density modified and sufficient to build an incomplete initial model of RexT in Phenix AutoBuild[74,75]. Using this model, we performed iterative rounds of refinement in Phenix and manually

extended the partial model to include protein sidechains, and water molecules in COOT[76]. The resulting more complete model of SeMet RexT contains all protein residues except 1 and 104-112. Analysis of the Ramachandran statistics using the MolProbity program showed that 97.99%, 2.01%, and 0% of residues are in the favored, allowed, and disallowed regions, respectively.

The SeMet-RexT structure was used to solve a higher, 1.95-Å resolution structure of native RexT by molecular replacement. The native RexT structure was used to solve the structure of the oxidized RexT (RexT_ox) to 2.16-Å resolution, by isomorphous replacement. The models of RexT and RexT_ox were refined similarly to that explained above for SeMet-RexT; using iterative rounds of positional refinement and individual B-factor refinement in Phenix, and model adjustment, and the addition of water, glycerol, and molecules in COOT[76]. For RexT_ox, the initial solution showed negative difference density in the region that included Cys40 and Cys41 in chain B. Correct modeling of the disulfide bond was attempted through several rounds of building and testing of different orientations until the model best matched the electron density. These structures contain all protein residues except 106–112. Analysis of the Ramachandran statistics using the MolProbity program showed that 99.02%, 0.98%, and 0% of residues are in the favored, allowed, and disallowed regions, for RexT, 96.98%, 3.02%, and 0% of residues are in the favored, allowed, and disallowed regions, for RexT_ox.

For SeMet-RexT, RexT, RexT_ox, the data statistics are summarized in Table 1. All structures were refined using identical $R_{free}$ test sets (10% originating from the SeMet dataset) to evaluate the progression of refinement and simulated annealing composite omit maps were generated to verify the structures. Crystallography software packages were compiled by SBGrid[77].

**Creation of molecular variants of RexT.** Molecular variants of RexT were generated using the QuikChange™ Lightning Site-Directed Mutagenesis Kit (Agilent) and the primers (Integrated DNA Technologies) listed in Supplementary Table 3. All plasmid sequences were confirmed by sanger DNA sequencing (Genewiz). Conditions for overexpression and purification of each RexT variant were identical to that described above for wild-type RexT.

**Electrophoretic mobility shift assay (EMSA) for DNA-binding.** To confirm that the tag-free RexT construct used in this study behaves similarly to the His-tagged RexT (no cleavable sequence) from the previous study[17] and to check the DNA-binding ability of RexT variants, the 325 base-pair intergenic sequence between rexT and trxA2 (see Supplementary Table 1) was commercially synthesized (GenScript) and amplified by PCR using the primers: 5'-GCTTGCTAA-CAATCGCACAATCTCC-3' and 5'-TCGCTAGCAACTTCATCCACAACC-3'. This DNA fragment was extracted and purified following electrophoresis on a 1.5% agarose gel (Invitrogen).

For the electrophoretic mobility shift assay (EMSA) used to probe DNA-binding, previous literature protocols were followed albeit with slight modifications[17]. In brief, 20 nM of the amplified DNA fragment was incubated with 25, 50, 100 nM dimeric RexT or its variants at 21 °C in binding buffer (20 mM HEPES (pH 8.0), 100 mM NaCl, 10 mM MgCl₂, 10% v/v glycerol, and 1 mM DTT) for 30 min. To evaluate the effect of possible metal or arsenite binding on DNA binding, 200 nM, 500 nM, and 1000 nM of ZnCl₂, CdCl₂, and sodium arsenite were added to the 100 nM dimeric RexT sample. Samples were analyzed by electrophoresis on 7.5% native polyacrylamide gels (Bio-Rad) at 45 V for 120 min using a Tris-Borate-EDTA (TBE) buffer. DNA fragments were visualized by staining with GelRed® nucleic acid gel stain (Biotium).

**Fluorescence anisotropy experiments.** The DNA duplex fluorescence probe (Supplementary Table 1) was synthesized with one 5'-end labeled with 6-carboxyfluorescein (5' 6-FAM, IDT). Different concentrations of RexT and its variants prepared by two-fold serial dilution to achieve a final dilution of 15.625 nM to 32 μM (up to 64 μM for R26A and K50A) and were incubated with 100 nM DNA fluorescence probe in binding buffer (20 mM HEPES (pH 8.0), 100 mM NaCl, 10 mM MgCl₂, 10% v/v glycerol, and 1 mM DTT) for 15 min at room temperature. For $Cd^{2+}$ treated RexT, 2.5 equivalent of CdCl₂ were added to the highest concentration of protein stock and then diluted as described above. Fluorescence anisotropy experiment was performed in 96-well plates on a PHERAstar reader (BMG Labtech) with 485 nm excitation and 520 nm emission filters. Fluorescence anisotropy was calculated using equation (1).

$$A = \frac{I_{||} - I_{\perp}}{I_{||} + 2I_{\perp}} \qquad (1)$$

Fluorescence anisotropy values were further normalized using equation (2)

$$A_{normalized} = (A_{obs} - A_0)/(A_{max} - A_0) \qquad (2)$$

as previously reported[78], where $A_{obs}$ is the measured fluorescence anisotropy at a given protein concentration, $A_{max}$ is the maximum fluorescence anisotropy obtained in the plateau region of the curve, and $A_0$ is the fluorescence anisotropy of the DNA probe only. For R26A and K50A variants, the titration curves don't reach plateau values and the $A_{max}$ is calculated by adding the change of wild-type RexT ($A_{max}$-$A_0$) to the corresponding $A_0$ of the variants. The normalized data were analyzed in Prism GraphPad by fitting into a simple, one-step DNA-RexT binding

model, shown in equation (3).

$$Y = \frac{A_{max} * X}{K_d + X} \qquad (3)$$

**H₂O₂ consumption assay**. To compare how $H_2O_2$ was consumed by RexT and its molecular variants, a modified version of a previously published ferrous oxidation of xylenol orange (FOX) assay was used[9]. RexT or its variants were first buffer-exchanged into the reaction buffer (100 mM sodium phosphate, pH 7.4) with a desalting column (Bio-Spin P-6 Gel Columns, Bio-Rad) to remove the DTT present in the storage buffer right before being added into the reaction mixture at a final concentration of 100 μM. To initiate the reaction, $H_2O_2$ was added to the mixture to a final concentration of 200 μM. 20 μL aliquots of the mixture were taken after 10, 20, 30, 40, 60, 90, 150, 300, and 500 s, mixed with 980 μL of the FOX agent (100 μM xylenol orange, 250 μM ammonium ferrous sulfate, 100 mM sorbitol and 25 mM $H_2SO_4$), and incubated for 30 min at room temperature in the dark. At the end of the reaction, the absorbance at 560 nm was measured on a 96-well plate reader (Epoch 2 microplate spectrophotometer, BioTek). Negative controls were included by using the reaction buffer, or 1.3 mg/ml bovine serum albumin (BSA) in the place of RexT. The concentration of $H_2O_2$ was determined by an $H_2O_2$ standard curve.

**Statistics and reproducibility**. All biochemical experiments were typically performed in independent triplicates, with two or three different batches of protein or DNA samples. For $H_2O_2$ consumption assays and fluorescence anisotropy experiments, the results are presented as the mean ± SD with individual data points overlayed in the same graph. For EMSA, mass spectrometry, and CD experiments, representative images and graphs are presented. Data for protein X-ray crystallography were collected on different crystals produced from different batches of protein samples with similar results.

**Mass spectrometry experiments**. Liquid chromatography-mass spectrometry (LC-MS) analysis was performed for RexT and its variants in their as-purified, $H_2O_2$-treated, and $H_2O_2$/dimedone-treated forms. As-purified samples were prepared by diluting protein stock solutions with water to a final concentration of 5 μM. $H_2O_2$-treated samples were prepared by mixing 10 μM RexT or its variants with 400 μM $H_2O_2$ at a 1:1 ratio and incubated at 30 °C for 2 h. For dimedone modification, protein samples with 10 mM dimedone added (Tokyo Chemical Industry) were mixed with 400 μM $H_2O_2$ at a 1:1 ratio and incubated at 30 °C for 14 h. The samples were then injected into LC-MS for further analysis.

LC-MS experiments were performed on an Agilent G6545A liquid chromatography/quadrupole-time of flight (LC/Q-TOF) equipped with a dual AJS ESI source and an Agilent 1290 Infinity series diode array detector, autosampler, and binary pump. An Aeris WIDEPORE C4 column (2.1 × 50 mm, 3.6 μm, 200 Å) (Phenomenex) was used for sample separations, with Solvent A = water with 0.1% formic acid and Solvent B = 95% acetonitrile, 5% water, and 0.1% formic acid. The employed chromatographic method used (i) 5% Solvent B 0–2 min, (ii) a linear gradient to 90% Solvent B over 5 min, and (iii) a final step that used 90% Solvent B for 1 min. 5.0 μL injections were made for each sample and the column was run at 0.3 mL/min. MS data were analyzed and tabulated in Agilent MassHunter BioConfirm.

**Circular dichroism (CD) experiments**. Stock solutions of wild-type RexT were diluted to 10 μM using a buffer containing 50 mM HEPES (pH = 8.0) and 150 mM NaCl. $H_2O_2$ was added into the protein solution with a final concentration of 1:1 or 2:1 to protein concentration. This mixture was incubated at room temperature for 5, 15, and 30 min and the CD spectrum of each sample was taken subsequently. For each CD measurement, 400 μL of the diluted RexT -$H_2O_2$ solution was transferred into a 10 mm quartz cuvette (Hellma). The CD spectra were recorded using a Jasco J-1500 CD spectrometer with 0.1 nm data pitch and 20 nm/min scan speeds. The baseline was measured with the same buffer used to do dilution (50 mM HEPES (pH = 8.0) and 150 mM NaCl). Each sample spectrum is an average of five cumulative spectra.

**Bioinformatics analysis**. The protein family classification for RexT was analyzed through the InterPro database[79] by searching the full amino acid sequence of RexT. The 6-99 amino acid region was identified as the HTH ArsR-type DNA-binding domain (IPR001845), which was shared with typical members of the ArsR-SmtB family. A sequence similarity network (SSN) for representative sequences of the ArsR-SmtB family was generated by EFI-Enzyme Similarity Tool[80,81]. The resultant SSN contained 20616 UniRef50[82] sequences, or proteins that share 50% sequence identity in the UniProt database. These sequences were analyzed by Cytoscape[83] at a cut-off value of e$^{-20}$. Biochemically and/or structurally characterized proteins were manually highlighted in the network. Key amino acid sequences from the network were aligned by Clustal W[84].

To explore the cluster that contains RexT in more detail, 7085 of the original amino acid sequences were analyzed using the EFI-Genome Neighborhood Tool[81,85,86]. The genome neighborhood diagrams show that only 105 out of the 7085 clustered proteins are adjacent to a thioredoxin gene. Sequence logos of these

105 proteins were generated by WebLogo 3[87]. A phylogenetic tree of proteins from three orders of cyanobacteria was built by MEGA X[88].

**Reporting summary**. Further information on research design is available in the Nature Research Reporting Summary linked to this article.

## Data availability
Protein coordinates and structure factors have been submitted to the Protein Data Bank under accession codes 7TXO (SeMet-RexT), 7TXN (Reduced RexT), and 7TXM (Oxidized RexT). The source data underlying Figs. 2, 3, and 5 and Supplementary Fig. 3 are provided as a source data file. Other data are available in the Supplementary Information and from the corresponding authors upon reasonable request.

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

## Acknowledgements

This material is based upon work supported by the U.S. Department of Energy, Office of Science, Office of Basic Energy Sciences, under Award Number DE-SC0021240 (J.B.R), the Searle Scholars Program (J.B.R), and the University of Michigan Chemistry Department Summer Undergraduate Research Program (R.C). In addition, this research used resources of the Advanced Photon Source, a U.S. Department of Energy (DOE) Office of Science User Facility operated for the DOE Office of Science by Argonne National Laboratory under Contract No. DE-AC02-06CH11357. Use of the LS-CAT Sector 21 was supported by the Michigan Economic Development Corporation and the Michigan Technology Tri-Corridor (Grant 085P1000817). The authors thank Lindsey Backman for helpful discussion and feedback on this manuscript and the beamline scientists at LS-CAT, especially Dr. Zdzislaw Wawrzak, for assistance. The authors also thank Dr. Aaron Robida of the Center for Chemical Genomics at the University of Michigan, Ann Arbor for assistance with fluorescence anisotropy experiments.

## Author contributions

B.L., M.J., J.L., J.T., and J.B.R. contributed to the design of the experiments and wrote the manuscript. B.L. and M.J. expressed and purified all native and SeMet proteins used in this work. B.L. performed all $H_2O_2$ consumption, fluorescence anisotropy, and EMSA experiments. B.L. performed all crystal optimization, crystallographic data collection, and solved the three X-ray crystallographic structures determined in this work. B.L. conducted sequence similarity network and phylogenetic analysis. J.L. performed mass spectrometry experiments. J.T. performed the CD experiments. R.C. performed initial crystallographic screening and optimization experiments.

## Competing interests

The authors declare no competing interests.
