## [Peer Review File · Communications Biology]

Reviewers' comments:

Reviewer #1 (Remarks to the Author):

The authors present a structural and chemical characterization of RexT, an ArsR transcriptional repressor that responds to H₂O₂ in the cyanobacterium *Nostoc* sp PCC 7120. This study provides new structural and chemical reactivity data, as well as a comprehensive analysis of SSN from ArsR family proteins. The authors report the first crystal structure of the reduced state of the only sensor in this family that has been shown to sense ROS. Additionally, they report a H₂O₂ soaked crystal structure where one of the Cys sites forms a vicinal disulfide and the functional role of these two cysteines is further validated very precisely by FOX assays and site-directed mutagenesis. Further analysis of the crystal structure allows the authors to identify key residues implicated in reactivity with H₂O₂. Finally, they present a comprehensive analysis of SSN from ArsR family that suggests that the vicinal disulfide is exclusive from the RexT-like proteins in the Nostocales order.

This is a beautifully conducted study and presents very compelling data from the unique H₂O₂-sensing ArsR transcriptional repressor. The results and analysis of the data are of high quality. The findings presented in this manuscript should be of significant interest to the bioinorganic and structural biology community, particularly the redox community.

My main criticism of the work is that in several places it seems to me that the authors included unnecessary overinterpretation of the data that often contradicts what has been reported in this family of proteins (some of this work is even commented by the authors along the manuscript). Thus, I have some suggestions that in my opinion would improve the quality of the manuscript and prevent confusion of the non-specialist and frustration of the specialist:

1) Suggesting that ArsR proteins an "obligatory" metalloregulatory scaffold is just not right, there are to date at least 20 published papers describing members of the family that sense other inducers that are not metal/metalloid ions (the authors are pointed to Roy, *Metallogenics*, 2018 and Capdevila, *Essays in Biochemistry*, 2017; for a more thorough discussion of ArsR as a family that has a significant number of members that do not respond to metal/metalloid ions). The idea that ArsR are exclusively metalloregulators is present in the title, which I wouldn't suggest changing since for historical reasons the ArsR family of proteins has been a metalloregulatory family. However, I think this idea is overemphasized in the introduction and seems to be suggesting that the novelty of the paper is characterizing a nonmetalloregulator ArsR family protein. In my opinion, this is not the highlight of the paper since this has been done with BigR, HlyU, NolR, YgaV, SqrR, PigS, CyeR, PagR, Rv2034, Rv0081 and SrnR (this is a Ni-responsive regulator however it does not bind the metal). The list is even longer is the authors consider the closely related MarR-DUF family with HypR, YodV and QsrR. This paper is interesting as it is: the first structural and chemical characterization of a ROS sensor from the ArsR family. Thus, I suggest:

- the two sentences in the last 4 lines of page 3 are rewritten more scholarly, trying not to omit the research from all the non-metal sensors in this family of proteins.

- Lines 13-15 from page 4 can be removed or rewritten

- In the last sentence of the introduction, "metal binding protein can be reconfigured to showcase peroxidatic activity and respond to oxidative stress" should be rewritten not to imply that these proteins had a common ancestor that binds metals (which would contradict Roy, *Metallogenics*, 2018). It could be replaced by "a common ancestor protein can evolve to either showcase peroxidatic activity and respond to oxidative stress or sense metal ions" or something along those lines.

2) The section about the identification of RexT structural features involved in DNA binding has some interpretations that in my opinion should be reconsidered. It should be noted RexT in the reduced state has a quite unusual DNA binding competent conformation that is shared with NolR (which is probably the single member of the ArsR family that does not change conformation upon DNA binding, the authors are pointed to Arunkumar, *PNAS*, 2009 for a more detailed discussion). This fact allows the authors to speculate on the possible interaction with DNA by simply analyzing the reduced state. While the observation of the putative DNA binding residues along alpha 4 is scholarly adequate and the validation with EMSA assays provides further validation, the speculation of the Cys forming contact with phosphate is not. Particularly, since RexT as well as most ArsR with alpha3 inducer recognition site (like canonical ArsR) are expected to harness an allosteric connection between the Cys site and the DNA binding site. Thus, the decreased affinity of the Cys mutants is likely due to a simple structural perturbation of the allosteric site and the

provocative hypothesis that these allosteric residues participate in direct contact with the DNA is in my opinion not supported by the data.

3) In the section "A vicinal disulfide-bond mediated conformational change in RexT" the authors introduce a partially oxidized conformation obtained by soaking. As the authors point out the oxidation is not complete due to the limitations of crystal packing. A similar effect is also present in the conformation itself. Thus, the interpretation of the conformational changes should be done with caution because the conformational changes in solution are probably much more extended than the ones captured by this crystal structure. While there is a lot of valuable information that can be extracted from this structure, the analysis of the b-factors in the interface that is affected by packing seems too much of a stretch. I would suggest that the authors focus on the novelty of the proximal disulfide beautifully crystalized and leave the speculation about dynamics (the authors are pointed to this review that discusses possible pitfall of using b factors to predict dynamics in solution without other computational or experimental techniques: Sun, Chem. Rev., 2019) for the discussion or for further investigation of the dynamics in solution or in silico. The authors should note that it has been established in most wHTH motifs that the b-wings are very flexible in solution and the Asn77 is likely visiting a wide arrange of conformations in solution in both, the oxidized and reduced states. Finally, the last paragraph of this section should be introduced at the beginning, so the reader has complete information about the degree of structural change observed.

Minor

a) Page 4 lines 9: I would suggest changing "identification as a target" for "identification as a potential target"

b) Page 5 first paragraph last line: the Cys residues found on the $\alpha 5$ helix of HlyU is actually not that far from the C terminal Cys in RexT that is not functional. I would suggest removing the reference to the $\alpha 5$ -Cys from this sentence

Overall, I am very enthusiastic about this manuscript, I hope the authors would find my comments useful to strengthen their paper.

Reviewer #2 (Remarks to the Author):

Bin Li et al. presents an interesting and well-written descriptive structural and qualitative biochemical study on the molecular details of RexT as an H₂O₂ sensor. The study has several plausible interpretations or possible conclusions but misses at several points quantitative data to back-up and explain the final conclusions and interpretations of this study.

Authors state that (i) it is unknown how RexT binds DNA, which residues RexT uses to sense and react to H₂O₂, and how formation of a disulfide bond results in de-repression of transcription. That (ii) it is unclear how RexT, an ArsR-SmtB type transcriptional regulator, is built to respond to H₂O₂ stress, and rather than changes in metal ion homeostasis.

Like already forementioned, this study needs quantitative data and therefore some additional experiments are required.

Major experiments to add:

1. To investigate the metal-independent regulatory mechanism by which RexT senses and propagates a response to H₂O₂, authors used X-ray crystallography and site-directed mutagenesis to establish the molecular details of the RexT-based mode of DNA regulation. (i) Authors should show that RexT is not binding to As(III), Zn²⁺, Cd²⁺ in a quantitative study. (ii) Now it is an assumptions based on structural comparisons. Further, it is not clear why authors decided to add chlorides in the structure. Is there additional experimental evidence? This needs to be clarified.
2. Authors should determine the second order rate constant of RexT and its Cys mutants for H₂O₂, and discuss the results in the light of other transcription factors (fe OxyR). This is a relatively easy stopped-flow experiment in which authors should run experiments under pseudo first order conditions with increasing H₂O₂ concentrations.

3. 'Here, reminiscent of that observed with NoIR variants R31A and Q56A, we found substitution of these residues with an Ala residue weakens the ability of RexT to interact with DNA,...'. Authors should measure K_d using fluorescence polarization experiments with fluorescent labeled oligonucleotides for the DNA binding-site. This would give this study a more quantitative value, which currently is missing.

4. Fig. 5 D,E,F,G,H: The response to H₂O₂ is relatively slow compared to how other H₂O₂ transcription factors are responding. What was used as positive control and negative control in this assay? The trace lines for the controls should be added.

5. 'Furthermore, the larger impairment of the C41S variant to consume H₂O₂, suggests it functions as the peroxidatic cysteine residue'. Authors should show the formation of a sulfenic acid on C41. This can easily be determined with mass spec in an experiment in the presence of dimedone and H₂O₂.

Minor:

-It is a pity that no structure with H₂O₂ could be presented but I know that this is not always possible and difficult to obtain. Have authors considered to look at a different sigma level to see whether potential oxygens are present at the suggested hydrogen bonding location with R61? The distance between the oxygens in glycerol (Fig 5B) in interaction with R61 is larger than for 2 oxygens of H₂O₂ (see Prx and OxyR studies). Analysis of the oxidized structure shows a green density close to C41 and N77 in one of the molecules, the other molecule is forming a disulfide. Have authors considered to add H₂O₂ in this density difference map?

-Comparing Fig 5A and B, it looks that the position of the O δ 1 and N δ 2 of N77 is not the same. Please double check. I also noticed that in the SeMet structure the O δ 1 and N δ 2 of N77 are wrongly positioned. This needs to be corrected.

-Fig 5 B: Add the electron density.

-'We hypothesize that the magnitude of difference between the rates of H₂O₂ consumption in these proteins is attributable to the different levels of cellular H₂O₂ in the organisms that use these regulators.' I would rather go for a molecular explanation in which you compare the location of the two oxygens of H₂O₂ and their electrophilicity in terms of receiving a nucleophilic attack of the sulfur of C41. You could use Prdx (Hall et al. 2010) and OxyR for comparison. This might explain a relative lower second order rate constant (See major point 2).

-FigS2A: Why are the Cys in RexT not boxed?

-FigS2D: 'As' in this figure should be As(III) for arsenite.

-FigS5 ABCD: Electron density is not clear. Please use other colors and only show the meshed area with smaller mesh thickness. It is hard to get the required info from this figure.

-'Coupled with the wild-type levels of H₂O₂ consumed by the C105S RexT variant, we conclude that Cys105 is not involved in the RexT- mediated oxidative stress response'. Can this statement be backed-up with a CD study showing no incredible structural changes after the addition of H₂O₂ to WT?

-'Adjacent Cys residues are rarely used as metal-binding ligands'. Mention the examples that are known – see authors own supplementary material and add refs.

-Fig S2: Indication of the N- and C-terminal is not clear (arrows in different colors are confusing).

-Fig S5E should be based on experimental data. Detailed info on how the overlay with the DNA-bound NoIR structure (PDB: 4ON0) has been performed is missing here.

-Abstract: replace 'repurposed' by 'evolved'

-Title: Is much too broad. Replace 'in photosynthetic organisms' is too general. Change into 'cyanobacteria'. Further, 'if there is no metalloregulatory scaffold' as authors claim in their results, how can it be 'repurposed'?? Title needs to be changed.

-'the redox status of the cell' does not exist. Please rephrase.

-Fig 7. Density for H₂O₂ in the structure? (see the previous comment).

-Page 7: '...residues found in found in Corynebacterium glutamicum..'

Reviewer #1 (Remarks to the Author):

The authors present a structural and chemical characterization of RexT, an ArsR transcriptional repressor that responds to H₂O₂ in the cyanobacterium *Nostoc* sp PCC 7120. This study provides new structural and chemical reactivity data, as well as a comprehensive analysis of SSN from ArsR family proteins. The authors report the first crystal structure of the reduced state of the only sensor in this family that has been shown to sense ROS. Additionally, they report a H₂O₂ soaked crystal structure where one of the Cys sites forms a vicinal disulfide and the functional role of these two cysteines is further validated very precisely by FOX assays and site-directed mutagenesis. Further analysis of the crystal structure allows the authors to identify key residues implicated in reactivity with H₂O₂. Finally, they present a comprehensive analysis of SSN from ArsR family that suggests that the vicinal disulfide is exclusive from the RexT-like proteins in the Nostocales order.

This is a beautifully conducted study and presents very compelling data from the unique H₂O₂-sensing ArsR transcriptional repressor. The results and analysis of the data are of high quality. The findings presented in this manuscript should be of significant interest to the bioinorganic and structural biology community, particularly the redox community. My main criticism of the work is that in several places it seems to me that the authors included unnecessary overinterpretation of the data that often contradicts what has been reported in this family of proteins (some of this work is even commented by the authors along the manuscript). Thus, I have some suggestions that in my opinion would improve the quality of the manuscript and prevent confusion of the non-specialist and frustration of the specialist:

We thank Reviewer 1 for this positive assessment of our work, and we anticipate that that you will find the revised manuscript to be even more rigorous in the science and overall, more impactful to the community.

1) Suggesting that ArsR proteins an “obligatory” metalloregulatory scaffold is just not right, there are to date at least 20 published papers describing members of the family that sense other inducers that are not metal/metalloid ions (the authors are pointed to Roy, *Metallomics*, 2018 and Capdevila, *Essays in Biochemistry*, 2017; for a more thorough discussion of ArsR as a family that has a significant number of members that do not respond to metal/metalloid ions). The idea that ArsR are exclusively metalloregulators is present in the title, which I wouldn't suggest changing since for historical reasons the ArsR family of proteins has been a metalloregulatory family. However, I think this idea is overemphasized in the introduction and seems to be suggesting that the novelty of the paper is characterizing a nonmetalloregulator ArsR family protein. In my opinion, this is not the highlight of the paper since this has been done with BigR, HlyU, NoIR, YgaV, SqrR, PigS, CyeR, PagR, Rv2034, Rv0081 and SrnR (this is a Ni-responsive regulator however it does not bind the metal). The list is even longer is the authors consider the closely related MarR-DUF family with HypR, YodV and QsrR. This paper is interesting as it is: the first structural and chemical characterization of a ROS sensor from the ArsR family. Thus, I suggest:

- the two sentences in the last 4 lines of page 3 are rewritten more scholarly, trying not to omit the research from all the non-metal sensors in this family of proteins.

*We thank Reviewer 1 for this suggestion. We have rewritten this section of the manuscript to read: “This family was named after the founding members, *E. coli* arsenic and antimony regulatory protein (*EcArsR*) and *Synechococcus elongatus* PCC 7942 Zn²⁺-dependent regulatory protein*

(SeSmtB), and many of its members are recognized as metal-responsive transcriptional regulators (Fig. 1A). However, there are also members of this protein family that deviate from this metal-sensing role and instead have been shown to sense ROS or reactive sulfur species...” We have also added in the two suggested references to the manuscript.

- Lines 13-15 from page 4 can be removed or rewritten

These lines have been deleted from the manuscript

- In the last sentence of the introduction, “metal binding protein can be reconfigured to showcase peroxidatic activity and respond to oxidative stress” should be rewritten not to imply that these proteins had a common ancestor that binds metals (which would contradict Roy, Metallomics, 2018). It could be replaced by “a common ancestor protein can evolve to either showcase peroxidatic activity and respond to oxidative stress or sense metal ions” or something along those lines.

As suggested, the text has been updated to read: “...a common ancestor protein can evolve to showcase peroxidatic activity and respond to oxidative stress or sense metal ions”

2) The section about the identification of RexT structural features involved in DNA binding has some interpretations that in my opinion should be reconsidered. It should be noted RexT in the reduced state has a quite unusual DNA binding competent conformation that is shared with NolR (which is probably the single member of the ArsR family that does not change conformation upon DNA binding, the authors are pointed to Arunkumar, PNAS, 2009 for a more detailed discussion). This fact allows the authors to speculate on the possible interaction with DNA by simply analyzing the reduced state. While the observation of the putative DNA binding residues along alpha 4 is scholarly adequate and the validation with EMSA assays provides further validation, the speculation of the Cys forming contact with phosphate is not. Particularly, since RexT as well as most ArsR with alpha3 inducer recognition site (like canonical ArsR) are expected to harness an allosteric connection between the Cys site and the DNA binding site. Thus, the decreased affinity of the Cys mutants is likely due to a simple structural perturbation of the allosteric site and the provocative hypothesis that these allosteric residues participate in direct contact with the DNA is in my opinion not supported by the data.

We have added fluorescence anisotropy data for wild-type RexT binding to DNA and each of the variants from this section (R26A, K50A, C40S, and C41S). The text has been rewritten to include these results and they can be found in Figure 2E. We have also removed the discussion about the interaction between the Cys residues and DNA phosphates. Last, we have noted that NolR has an unusual DNA binding orientation that is similar to RexT.

3) In the section “A vicinal disulfide-bond mediated conformational change in RexT” the authors introduce a partially oxidized conformation obtained by soaking. As the authors point out the oxidation is not complete due to the limitations of crystal packing. A similar effect is also present in the conformation itself. Thus, the interpretation of the conformational changes should be done with caution because the conformational changes in solution are probably much more extended than the ones captured by this crystal structure. While there is a lot of valuable information that can be extracted from this structure, the analysis of the b-factors in the interface that is affected by packing seems too much of a stretch. I would suggest that the authors focus on the novelty of the proximal disulfide beautifully crystalized and leave the speculation about dynamics (the authors are pointed to this review that discusses possible pitfall of using b factors to predict dynamics in solution without other computational or experimental techniques: Sun, Chem. Rev.,

2019) for the discussion or for further investigation of the dynamics in solution or in silico. The authors should note that it has been established in most wHTH motifs that the b-wings are very flexible in solution and the Asn77 is likely visiting a wide arrange of conformations in solution in both, the oxidized and reduced states. Finally, the last paragraph of this section should be introduced at the beginning, so the reader has complete information about the degree of structural change observed.

As suggested, the last part of this section has been moved up to the beginning of the paragraph. "In chain A of the RexT dimer, a slight orientation difference is observed for both the Cys40 and Cys41 sidechains relative to the reduced structure."

In addition, the B factor description has been substantially truncated.

Minor

a) Page 4 lines 9: I would suggest changing "identification as a target" for "identification as a potential target"

The text has been updated as suggested.

b) Page 5 first paragraph last line: the Cys residues found on the $\alpha 5$ helix of HlyU is actually not that far from the C terminal Cys in RexT that is not functional. I would suggest removing the reference to the $\alpha 5$ -Cys from this sentence.

The text has been updated as suggested.

Overall, I am very enthusiastic about this manuscript, I hope the authors would find my comments useful to strengthen their paper.

Reviewer #2 (Remarks to the Author):

Bin Li et al. presents an interesting and well-written descriptive structural and qualitative biochemical study on the molecular details of RexT as an H₂O₂ sensor. The study has several plausible interpretations or possible conclusions but misses at several points quantitative data to back-up and explain the final conclusions and interpretations of this study.

Authors state that (i) it is unknown how RexT binds DNA, which residues RexT uses to sense and react to H₂O₂, and how formation of a disulfide bond results in de-repression of transcription. That (ii) it is unclear how RexT, an ArsR-SmtB type transcriptional regulator, is built to respond to H₂O₂ stress, and rather than changes in metal ion homeostasis.

Like already forementioned, this study needs quantitative data and therefore some additional experiments are required.

We thank Reviewer 2 for this positive assessment of our work. Based on the inclusion of several additional experiments, we anticipate that that you will find the revised manuscript to contain the needed data to support our findings.

Major experiments to add:

1. To investigate the metal-independent regulatory mechanism by which RexT senses and propagates a response to H₂O₂, authors used X-ray crystallography and site-directed mutagenesis to establish the molecular details of the RexT-based mode of DNA regulation. (i) Authors should show that RexT is not binding to As (III), Zn²⁺, Cd²⁺ in a quantitative study. (ii) Now it is an assumptions based on structural comparisons. Further, it is not clear why authors decided to add chlorides in the structure. Is there additional experimental evidence? This needs to be clarified.

We thank reviewer 2 for this comment. We have now added an additional EMSA experiment that was performed in the presence of 1, 2.5, and 5 equivalents of As (III), Zn²⁺, Cd²⁺. In addition, we have added a fluorescence anisotropy experiment to the manuscript to show that Cd²⁺ does not significantly change the ability of RexT to interact with DNA. From these experiments, it was determined that the K_d with and without Cd²⁺ is comparable (1065 nM (no Cd²⁺) and 1027 nM (+Cd²⁺)). The text has been updated to reflect the addition of these experiments and a Supplementary Figure has been added to the manuscript (Fig. S3 and Table S3). Chloride was added to the structure because it is present in the crystallization buffer. It refines well in the density unlike a water molecule. We have added maps into Fig. S6B-D.

2. Authors should determine the second order rate constant of RexT and its Cys mutants for H₂O₂ and discuss the results in the light of other transcription factors (fe OxyR). This is a relatively easy stopped-flow experiment in which authors should run experiments under pseudo first order conditions with increasing H₂O₂ concentrations.

We thank reviewer 2 for this comment. However, we find that using these types of methods described for other transcription factors is not feasible for RexT as it lacks the needed Trp residue to perform this analysis. Whereas we understand that we could incorporate a Trp residue for such an experiment, creation of a RexT variant protein would require extensive characterization to make sure that we had not compromised the structure of function of the protein. For these reasons, we assert that such experiments are outside the scope of this manuscript.

3. 'Here, reminiscent of that observed with NoIR variants R31A and Q56A, we found substitution of these residues with an Ala residue weakens the ability of RexT to interact with DNA,...'. Authors should measure K_d using fluorescence polarization experiments with fluorescent labeled oligonucleotides for the DNA binding-site. This would give this study a more quantitative value, which currently is missing.

We added fluorescence anisotropy studies to this manuscript to show that R26A and K50A variants of RexT decrease the affinity for DNA binding. This data is now included in Fig. 2E and Table S3.

4. Fig. 5 D, E, F, G, H: The response to H₂O₂ is relatively slow compared to how other H₂O₂ transcription factors are responding. What was used as positive control and negative control in this assay? The trace lines for the controls should be added.

The control lines have now been added into Figure 2B. Namely, we added the background assay that does not contain RexT as a control. We also tested a control reaction that contained BSA rather than RexT. Neither of these reactions showed consumption of H₂O₂. This assay was modeled after that previously published for OxyR and we used equivalent assay conditions to those previously published and referenced in this paper (reference 9). To make this latter point

clear, we added a statement to the text that reads “This assay has been previously described and used to show H₂O₂ consumption in OxyR. Using similar conditions to those described for OxyR...”

5. ‘Furthermore, the larger impairment of the C41S variant to consume H₂O₂, suggests it functions as the peroxidatic cysteine residue’. Authors should show the formation of a sulfenic acid on C41. This can easily be determined with mass spec in an experiment in the presence of dimedone and H₂O₂.

We thank reviewer 2 for this suggestion. We added mass spectrometry experiments into this work that show a disulfide bond is formed between Cys40 and Cys41 in RexT. Specifically, these results show that a disulfide bond is formed in wild-type RexT and C105S RexT. In contrast, both the C40S and C41S variants, as revealed by MS, do not form this bond. In addition, these experiments revealed that one equivalent of dimedone was incorporated in both wild-type and C41S RexT. In the C40S variant, we instead found that two equivalents were incorporated. In the C105S variant, no dimedone was incorporated. These results support our assessment that C41S behaves as the peroxidatic Cys residue. We hypothesize that we see incorporation of dimedone at the C105 position due to the reactive nature of Cys residues. We further suggest that disulfide bond formation between C40 and C41 is faster than dimedone incorporation, explaining why we don’t see it incorporated except for in the C40S variant. This data is now included as Fig. S7.

Minor

-It is a pity that no structure with H₂O₂ could be presented but I know that this is not always possible and difficult to obtain. Have authors considered to look at a different sigma level to see whether potential oxygens are present at the suggested hydrogen bonding location with R61? The distance between the oxygens in glycerol (Fig 5B) in interaction with R61 is larger than for 2 oxygens of H₂O₂ (see Prx and OxyR studies). Analysis of the oxidized structure shows a green density close to C41 and N77 in one of the molecules, the other molecule is forming a disulfide. Have authors considered to add H₂O₂ in this density difference map?

We thank Reviewer 2 for this suggestion. We have now modeled peroxide into this difference density, and it refines well. Extensive testing was also done to make sure that H₂O₂ refines better than a water molecule and also better than two water molecules in the density. The refined structure and maps are now included in Fig. 5A-B and Fig. S11E-F. The text has been updated to reflect these changes.

-Comparing Fig 5A and B, it looks that the position of the O δ 1 and N δ 2 of N77 is not the same. Please double check. I also noticed that in the SeMet structure the O δ 1 and N δ 2 of N77 are wrongly positioned. This needs to be corrected.

We have carefully checked the side chain positions in all of the structures.

-Fig 5 B: Add the electron density.

We have added a Supplementary Figure (Fig S11) that mirrors the panel in Figure 5B but also contains the electron density. Please note that the original 5B is now Fig. S11B. The maps for the chain that contains glycerol and the chain that contains H₂O₂ (new Fig. 5B) are both in Fig. S11.

-‘We hypothesize that the magnitude of difference between the rates of H₂O₂ consumption in these proteins is attributable to the different levels of cellular H₂O₂ in the organisms that use these regulators.’ I would rather go for a molecular explanation in which you compare the location

of the two oxygens of H₂O₂ and their electrophilicity in terms of receiving a nucleophilic attack of the sulfur of C41. You could use Prdx (Hall et al. 2010) and OxyR for comparison. This might explain a relative lower second order rate constant (See major point 2).

We have added additional text in this section that reads: “At the molecular level, it is also possible that the lower observed rate of H₂O₂ consumption in RexT is due to the different positioning of H₂O₂ in the active site relative to that seen in OxyR. In OxyR, the oxygen atom of H₂O₂ that is attacked by the nucleophilic Cys residue engages in multiple hydrogen bonding interactions with active site residues (Fig. 5C)⁹. These interactions presumably increase the electrophilicity of the oxygen atom and thereby facilitate the S_N2 displacement reaction. The equivalent oxygen atom of H₂O₂ observed in RexT does not engage in any such interactions (Fig. 5B).”

-FigS2A: Why are the Cys in RexT not boxed?

We have now added orange boxes around the Cys residues in RexT (Fig. S2A).

-FigS2D: ‘As’ in this figure should be As(III) for arsenite.

The figure has been updated as suggested.

-FigS5 ABCD: Electron density is not clear. Please use other colors and only show the meshed area with smaller mesh thickness. It is hard to get the required info from this figure.

We have updated panels A, B, C, and D in Fig. S5. We changed the 2Fo-Fc electron density maps to be shown in gray rather than blue. We also adjusted the mesh width as suggested.

-‘Coupled with the wild-type levels of H₂O₂ consumed by the C105S RexT variant, we conclude that Cys105 is not involved in the RexT- mediated oxidative stress response’. Can this statement be backed-up with a CD study showing no incredible structural changes after the addition of H₂O₂ to WT?

A CD study has been added to the manuscript (Fig. S8). The text now reads: “Based on this observation, the wild-type levels of H₂O₂ consumed by the C105S RexT variant, and a circular dichroism experiment that shows no large structural rearrangements following the addition of H₂O₂ to wild-type RexT, we conclude that Cys105 is not involved in the RexT-mediated oxidative stress response (Fig. 3C and Fig. S8).”

-‘Adjacent Cys residues are rarely used as metal-binding ligands’. Mention the examples that are known – see authors own supplementary material and add refs.

We have updated this statement as suggested and it now reads: “Intriguingly, despite the widespread use of Cys residues to coordinate metal ions in the ArsR-SmtB class of proteins, adjacent Cys residues are rarely used as metal-binding ligands³⁵. Likewise, although there are cases in this protein family where adjacent Cys residues are used to coordinate methyl-As(III) or As(III), all of the involved cysteine residues are found at the end of α 5 and the flexible region of the C-terminus.”

-Fig S2: Indication of the N- and C-terminal is not clear (arrows in different colors are confusing).

The figure has been updated so that the N- and C-terminus of the proteins are more clearly labeled.

-Fig S5E should be based on experimental data. Detailed info on how the overlay with the DNA-bound NoIR structure (PDB: 4ON0) has been performed is missing here.

We have added information into the legend about how the docking was performed (e.g. it was a simple alignment performed in PyMol). We have also added the above-mentioned fluorescence anisotropy experiments to show the importance of Arg26 and Lys50 to DNA binding.

-Abstract: replace 'repurposed' by 'evolved'

We have updated the text to read "... an interesting case where an ArsR-SmtB protein scaffold has been evolved to showcase peroxidatic activity and facilitate redox-based regulation."

-Title: Is much too broad. Replace 'in photosynthetic organisms' is too general. Change into 'cyanobacteria'. Further, 'if there is no metalloregulatory scaffold' as authors claim in their results, how can it be 'repurposed'?? Title needs to be changed.

The title has been changed to Structural and Mechanistic Basis for Redox Sensing by the Cyanobacterial Transcription Regulator RexT.

-'the redox status of the cell' does not exist. Please rephrase.

We have rephrased the sentence that contained this statement in the text to instead read: "In addition, many photosynthetic organisms use thioredoxin proteins to reduce disulfide bonds that are formed in response to ROS".

-Fig 7. Density for H₂O₂ in the structure? (see the previous comment).

See comments above.

-Page 7: '...residues found in found in Corynebacterium glutamicum..'

The text has been updated.

REVIEWERS' COMMENTS:

Reviewer #1 (Remarks to the Author):

All my comments have been properly addressed. I find indeed this version of the manuscript to be even more rigorous in the science and overall, more impactful to the community.

I have one minor concern about the new data that this version incorporated. The authors report a fluorescence anisotropy assay with a 35 bp oligo with a FAM label. The initial anisotropy is very low for a double-stranded oligo of this size which is typically around 0.1 (see Osman, NCB, 2019; Fakhoury, NAR, 2021, as examples). An easy explanation for this low anisotropy comes from the use of a regular FAM modification on the phosphate, which is a flexible linker to the DNA, instead of a fluoro-thymine which is more rigid and reports better on the tumbling time of the complex. As the authors also validate DNA binding with EMSA and they do see saturable curves with the reduced wild-type protein, my concern is mostly technical. However, I believe that the curves, particularly those of the mutants are overfitted. I would suggest the authors report a simpler fitting to a rectangular hyperbola and report in a table the initial and final anisotropy that comes from it, and not only a binding constant.

Reviewer #2 (Remarks to the Author):

Dear Authors,

Congrats with your work! I am happy with the adaptations. Excellent conducted research that resulted in a beautiful study of high level. Only 2 additional comments:

1. Fig2E and TableS3: The curves of the R26A and K50A mutants look very similar, while in Table S3 you report a different KD value. Further, I noticed that the information on how you determine the KD values is not mentioned in the M&M. Also, the number of replicates needs to be added in the figure legend.
2. Page 6: This finding also suggests that Cys40 is the resolving Cys, or the residue that reduces the formed sulfenic acid moiety to make a disulfide bond. Replace by: This finding also suggests that Cys40 is the resolving Cys, or the residue that reacts with the formed sulfenic acid moiety to make a disulfide bond.